# Tumor-derived extracellular vesicles regulate tumor-infiltrating regulatory T cells via the inhibitory immunoreceptor CD300a

Yuta Nakazawa[1,2], Nanako Nishiyama[1,2], Hitoshi Koizumi[1,2], Kazumasa Kanemaru[1,3], Chigusa Nakahashi-Oda[1,3]*, Akira Shibuya[1,3,4]*

[1]Department of Immunology, Faculty of Medicine, University of Tsukuba, Tsukuba, Japan; [2]Doctoral Program of Biomedical Sciences, Graduate School of Comprehensive Human Sciences, University of Tsukuba, Tsukuba, Japan; [3]R&D Center for Innovative Drug Discovery, University of Tsukuba, Tsukuba, Japan; [4]Life Science Center for Survival Dynamics, Tsukuba Advanced Research Alliance (TARA), University of Tsukuba, Tsukuba, Japan

*For correspondence:
chigusano@md.tsukuba.ac.jp
(CN-O);
ashibuya@md.tsukuba.ac.jp (AS)

**Competing interest:** The authors declare that no competing interests exist.

**Abstract** Although tumor-infiltrating regulatory T (Treg) cells play a pivotal role in tumor immunity, how Treg cell activation are regulated in tumor microenvironments remains unclear. Here, we found that mice deficient in the inhibitory immunoreceptor CD300a on their dendritic cells (DCs) have increased numbers of Treg cells in tumors and greater tumor growth compared with wild-type mice after transplantation of B16 melanoma. Pharmacological impairment of extracellular vesicle (EV) release decreased Treg cell numbers in CD300a-deficient mice. Coculture of DCs with tumor-derived EV (TEV) induced the internalization of CD300a and the incorporation of EVs into endosomes, in which CD300a inhibited TEV-mediated TLR3–TRIF signaling for activation of the IFN-β-Treg cells axis. We also show that higher expression of CD300A was associated with decreased tumor-infiltrating Treg cells and longer survival time in patients with melanoma. Our findings reveal the role of TEV and CD300a on DCs in Treg cell activation in the tumor microenvironment.

## Editor's evaluation

This report shows that the inhibitory immunoreceptor CD300a binding tumor-derived extracellular vesicles are incorporated into dendritic cells and inhibit their IFN-β production in tumor tissues. This results in suppressed activation of tumor-infiltrating regulatory T cells and consequently enhanced tumor immunity.

## Introduction

CD4[+] regulatory T (Treg) cells specifically expressing Foxp3 play an essential role for maintaining peripheral tolerance, preventing autoimmunity, and limiting chronic inflammatory diseases. Deficiency in Treg cells due to genetic inactivation of *Foxp3* or impaired induction of Treg cells after birth results in lethal autoinflammatory syndromes (*Kim et al., 2007*; *Ramsdell and Ziegler, 2014*). Treg cells are found at various tissues, including tumors, at various frequencies. Because tumor-infiltrating Treg cells suppress the activation of tumor antigen-specific CD8[+] T cells, a greater proportion of Treg cells to CD8[+] T cells among tumor-infiltrating lymphocytes is associated with poor prognosis in several

cancers (*Nishikawa and Sakaguchi, 2010*). Indeed, Treg cell depletion dramatically reduces tumor burden (*Klages et al., 2010*). Current clinical trials are evaluating strategies targeting receptors (CD25, CTLA-4, CCR4, OX40, and GITR) preferentially expressed on intratumoral Treg cells (*Nishikawa and Sakaguchi, 2010*; *Shitara and Nishikawa, 2018*). The migration of Treg cells and their activation and proliferation are regulated by chemoattractants (*Adeegbe and Nishikawa, 2013*; *Ondondo et al., 2013*) and cytokines such as TGF-β and IL-10 (*Hsu et al., 2015*; *Wan and Flavell, 2007*). However, how Treg cell activation and proliferation are regulated in the tumor microenvironments remains unclear.

Extracellular vesicles (EVs) are the particles released from the cell that are delimited by a lipid bilayer containing functional biomolecules (proteins, lipids, mRNAs, microRNAs, and DNA fragments) that can be transferred to other cells (*van Niel et al., 2018*; *Witwer and Théry, 2019*). More than 4000 trillion EVs are presumed to be in the blood of cancer patients (*Melo et al., 2015*) and EVs released from tumor cells (tumor-derived EVs [TEVs]) are emerging as critical messengers in tumor progression and metastasis (*Couto et al., 2018*; *Grange et al., 2011*; *Melo et al., 2015*; *Skog et al., 2008*). In tumor immunity, pleiotropic and deleterious role of TEV has been reported that, Fas ligand and PD-L1, the immunomodulatory molecules, on the surface of TEV induce apoptosis or suppression of activated T cells (*Andreola et al., 2002*; *Chen et al., 2018*) and TGF-β1 in TEV induces Treg cells (*Clayton et al., 2007*). TEV also upregulates PD-L1 expression on myeloid cells (*Fleming et al., 2019*). Furthermore, myeloid cells that capture microRNA within TEVs are altered to myeloid-derived suppressor cells and/or M2 macrophages and promote the malignant behavior of cancers (*Huber et al., 2018*; *Tian et al., 2019*; *Wang et al., 2018*; *Ying et al., 2016*). These intensive investigations have shown that TEVs play a key role in the suppression of antitumor immune responses (*Zebrowska et al., 2020*). However, how TEV regulates myeloid cell activation in tumor microenvironment is still incompletely understood.

The mouse CD300 family molecules, which are encoded by nine genes on chromosome 11, are expressed on myeloid cells including macrophages, dendritic cells (DCs), mast cells, and granulocytes and either activate or inhibit innate immune responses (*Borrego, 2013*; *Voss et al., 2015*). On the other hand, the human CD300 family consists of seven molecules encoded by genes located on chromosome 17 in a region syntenic to mouse chromosome 11 (*Clark et al., 2001*). CD300a, one of the CD300 molecules in mouse, contains two immunoreceptor tyrosine-based inhibitory motifs in its cytoplasmic portion. It mediates an inhibitory signal via SHP-1 and SHP-2 by binding to phosphatidylserine, which is exposed on the outer leaflet of the plasma membrane on apoptotic cells and activated mast cells under degranulation (*Nakahashi-Oda et al., 2012a*; *Wang et al., 2019*; *Yotsumoto et al., 2003*). Upon binding to phosphatidylserine, CD300a inhibits TLR4-mediated signaling in mast cells and DCs, which results in the suppression of cytokine and chemokine production and modulation of inflammatory immune responses (*Nakahashi-Oda et al., 2016*; *Nakahashi-Oda et al., 2012b*).

Here, we investigated the role of CD300a in tumor development and demonstrate that CD300a inhibits TEV-mediated interferon-β (IFN-β) production by DCs and suppresses the activation of tumor-infiltrating Treg cells and tumor development.

## Results

### CD300a on DCs enhances antitumor immunity

To address whether CD300a is involved in tumor immunity, wild-type and CD300a-deficient ($Cd300a^{-/-}$) mice were transplanted intradermally with B16 melanoma cells. The $Cd300a^{-/-}$ mice showed larger tumor volume and shorter survival than did wild-type mice (*Figure 1A and B*), indicating that CD300a suppresses the development of melanoma. In contrast, Rag-deficient ($Rag1^{-/-}$) and $Rag1^{-/-};Cd300a^{-/-}$ mice showed comparable levels of tumor development and survival after injection of B16 melanoma cells (*Figure 1C and D*). These results indicate that the suppressive effect of CD300a on melanoma development is dependent on the adaptive immune response. However, we also observed that CD300a was not expressed on tumor-infiltrating lymphocytes but was broadly expressed on myeloid cells, including populations of Ly6G$^+$ neutrophils, CD11c$^{+\sim high}$ DCs, and CD11c$^{low}$CD11b$^+$ macrophages (*Figure 1E*). These results suggest that CD300a expressed on myeloid cells suppresses melanoma development via adaptive immune responses. To identify the CD300a-expressing myeloid cell population that is involved in melanoma suppression, we used $Cd300a^{fl/fl};Itgax^{Cre}$ and $Cd300a^{fl/fl};Lyz2^{Cre}$ mice. $Cd300a^{fl/fl};Itgax^{Cre}$ mice expressed CD300a on Ly6G$^+$ cells and CD11c$^-$ cells, but not on CD11c$^{+\sim high}$

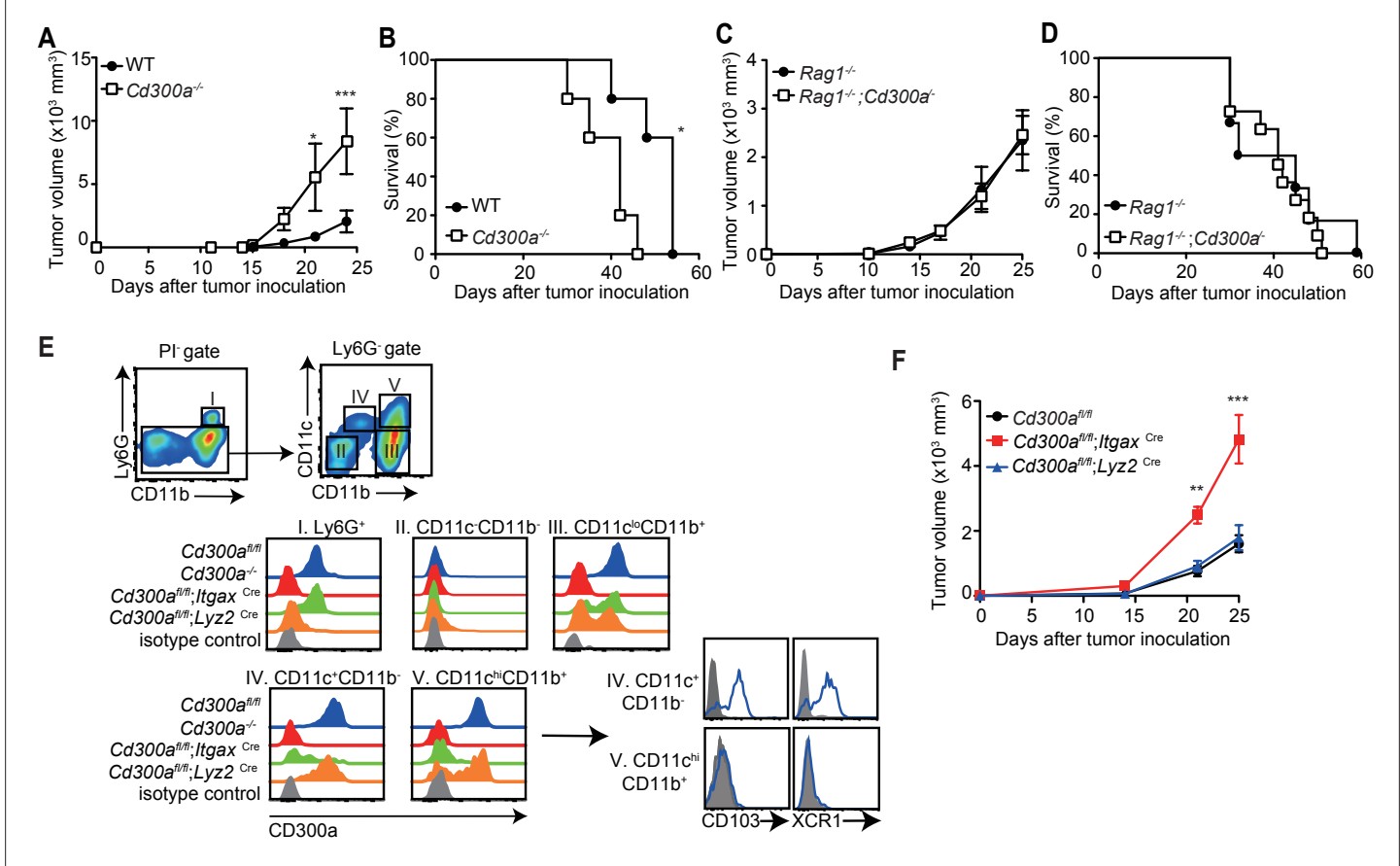

**Figure 1.** CD300a suppresses tumor growth. (**A–D**) Tumor growth or survival curves of wild-type (WT, $n = 5$ in **A** and **B**), $Cd300a^{-/-}$ ($n = 5$ in **A** and **B**), $Rag1^{-/-}$ ($n = 11$ in **C** and $n = 6$ in **D**), and $Rag1^{-/-};Cd300a^{-/-}$ ($n = 15$ in **C** and $n = 11$ in **D**) that were inoculated with $1 \times 10^5$ B16 melanoma cells on day 0. (**E**) CD300a expression on neutrophils (Ly6G$^+$), macrophages (Ly6G$^-$CD11c$^{lo}$CD11b$^+$), conventional type-1 DC (cDC1; Ly6G$^-$CD11c$^+$CD11b$^-$CD103$^+$XCR1$^+$), and cDC1 (Ly6G$^-$CD11c$^{+hi}$CD11b$^+$CD103$^-$XCR$^-$) isolated from B16 melanoma tissues of $Cd300a^{fl/fl}$, $Cd300a^{-/-}$, $Cd300a^{fl/fl};Itgax^{Cre}$, and $Cd300a^{fl/fl};Lys2^{Cre}$ mice prepared 14 days after inoculation. Data are representative of three mice. (**F**) Tumor growth of $Cd300a^{fl/fl}$ ($n = 7$), $Cd300a^{fl/fl};Itgax^{Cre}$ ($n = 13$), and $Cd300a^{fl/fl};Lys2^{Cre}$ mice ($n = 15$) that were inoculated with $1 \times 10^5$ B16 melanoma cells on day 0. Data are given as means ± standard error of the means (SEMs). **$p** < 0.01$ and ***$p < 0.001$. p values were obtained by using a two-way analysis of variance (ANOVA) followed by Bonferroni's post-test (**A, C, and F**) and the log-rank test (**B and D**). Data were pooled from two (**A– C and E**) or three (**D and F**) independent experiments.

The online version of this article includes the following figure supplement(s) for figure 1:

**Source data 1.** Source data for *Figure 1A-D and F*.

cells (*Figure 1E*). In contrast, $Cd300a^{fl/fl};Lyz2^{Cre}$ mice express CD300a on CD11c$^{+\sim high}$ cells and the subpopulation of CD11c$^{low}$ cells, but not on Ly6G$^+$ cells (*Figure 1E*). Although tumor growth was comparable between $Cd300a^{fl/fl};Lyz2^{Cre}$ and $Cd300a^{fl/fl}$ mice, $Cd300a^{fl/fl};Itgax^{Cre}$ mice showed greater tumor volume than did $Cd300a^{fl/fl}$ mice (*Figure 1F*). These data implicated CD300a on DCs, rather than on neutrophils or macrophages, in inducing the adaptive immune response to inhibit tumor development.

## CD300a regulates tumor-infiltrating Treg cells

Previous reports have demonstrated that the number of Treg cells in melanoma is correlated with accelerated tumor growth (*Mougiakakos et al., 2010*). In contrast, depletion of Treg cells leads to less melanoma growth. To elucidate how CD300a on DCs enhances the adaptive immune response against tumor development, we analyzed the population of tumor-infiltrating Treg cells by use of flow cytometry and immunohistochemistry. The Treg cell population was larger in the tumor, but not the draining lymph nodes, of $Cd300a^{-/-}$ mice compared with that of wild-type mice (*Figure 2A and B*). Likewise, $Cd300a^{fl/fl};Itgax^{Cre}$ mice showed a higher number of tumor-infiltrating Treg cells than did

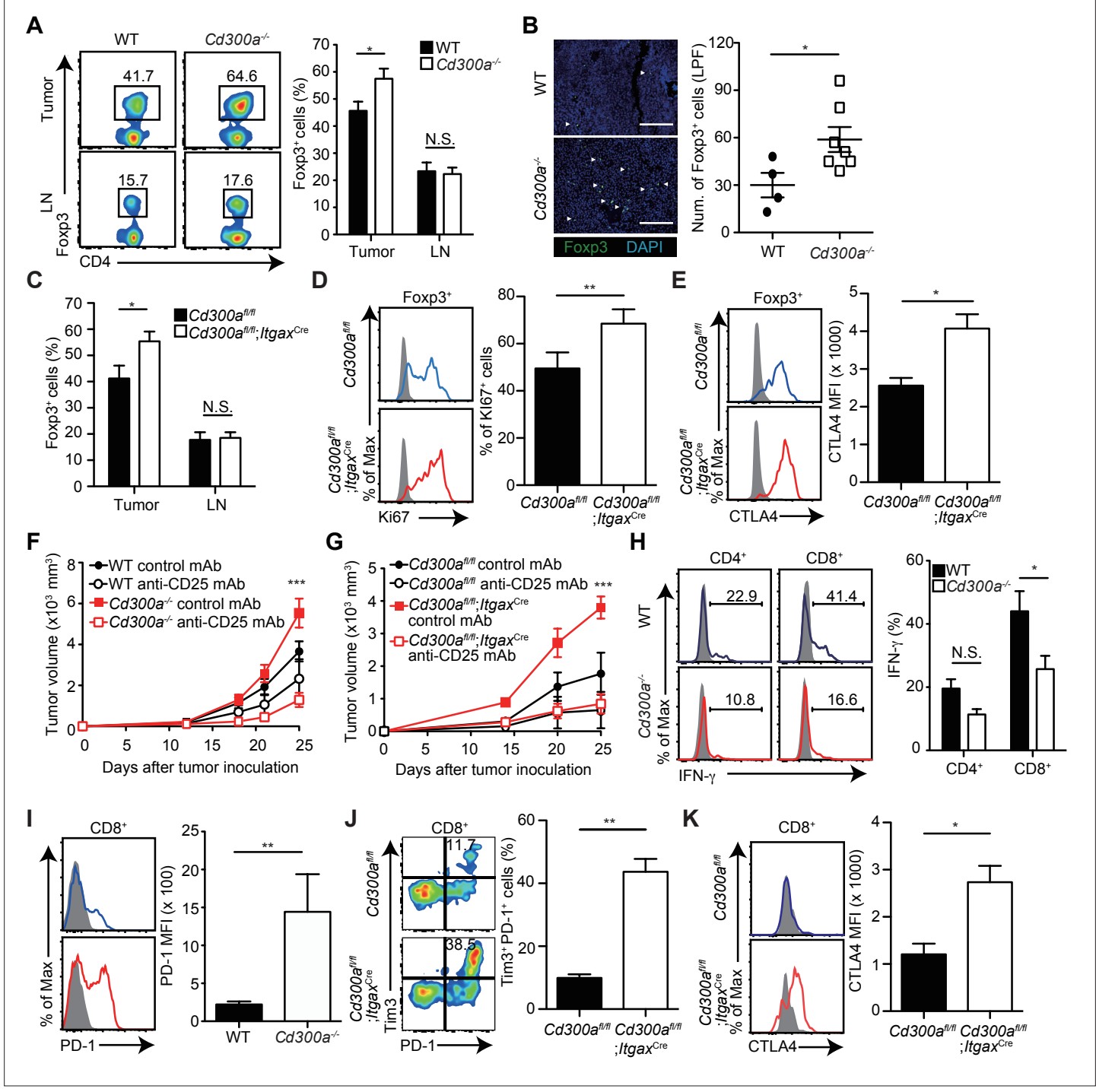

**Figure 2.** Tumor-infiltrating Treg cells are regulated by CD300a. Tumor tissues were harvested 3 weeks after B16 melanoma inoculation. (**A**) Representative flow cytometry plots of Treg cells in the tumor and draining lymph node (LN) (left). Numbers adjacent to outlined areas indicate the percentage of Foxp3+ (Treg) CD4+ cells. The frequencies of Foxp3+ cells among CD4+ T cells in both wild-type (WT, *n* = 7) and *Cd300a*−/− mice (*n* = 8) are shown (right). (**B**) Fluorescence microscopy of tumor sections from Foxp3-eGFP WT (*n* = 4) and *Cd300a*−/− (*n* = 7) mice, stained with an anti-GFP monoclonal antibody (green) and the DNA-binding dye 4',6-diamidino-2-phenylindole (DAPI; left). The number of Foxp3+ cells was quantified from for low-power fields (LPF) (right). White arrow shows Foxp3-positive cells. Scale bar, 200 μm. (**C**) Flow cytometric analysis of the frequencies of Foxp3+ cells among CD4+ T cells in the tumor (*n* = 11 in each group) and draining lymph node (*n* = 8 in each group) in *Cd300a*fl/fl and *Cd300a*fl/fl;*Itgax*Cre mice. Flow cytometric analysis of Ki67 (**D**) and CTLA-4 (**E**) expressions of Treg cells in the tumor in *Cd300a*fl/fl (*n* = 9 in D, *n* = 3 in E) and *Cd300a*fl/fl;*Itgax*Cre mice (*n* = 10 in D, *n* = 4 in E). Representative histogram (left), frequency (right), (**D**), and mean fluorescent intensity (MFI, right, **E**). (**F and G**) Tumor growth curve of WT (control mAb, *n* = 7; anti-CD25 mAb, *n* = 5), *Cd300a*−/− (control mAb, *n* = 8; anti-CD25 mAb, *n* = 6), *Cd300a*fl/fl (control mAb, *n* = 4; anti-CD25 mAb, *n*

*Figure 2 continued on next page*

*Figure 2 continued*

= 3), and *Cd300a*<sup>fl/fl</sup>;*Itgax*<sup>Cre</sup> (control mAb, n = 3; anti-CD25 mAb, n = 5) mice that were treated with an anti-CD25 mAb or a control antibody three times (days −6, −3, and 0) and then inoculated with B16 melanoma cells. (**H**) Representative histogram of IFN-$\gamma$ production from tumor-infiltrating T cells after phorbol 12-myristate 13-acetate (PMA) and ionomycin stimulation (left). The proportion of IFN-$\gamma^+$ cells is shown (right) (n = 6 in each group). (**I–K**) Flow cytometric analysis of the expressions of programmed cell death-1(PD-1) (**I**), PD-1 and Tim3 (**J**), and CTLA-4 (**K**) in CD8$^+$ T cells in the tumor of *Cd300a*<sup>fl/fl</sup> (n = 4 in I, n = 3 in J and K) and *Cd300a*<sup>fl/fl</sup>;*Itgax*<sup>Cre</sup> (n = 6 in I, n = 5 in J, n = 4 in K) mice. Representative histogram (left, **I and K**) or dot plots (left, **J**) and MFI (I and K, right) or frequency (J, right). Data are given as means ± standard error of the means (SEMs). N.S.: not significant. *p < 0.05, **p < 0.01, and ***p < 0.001. p values were obtained by using a two-way analysis of variance (ANOVA) followed by Bonferroni's post-test (**A, C, and H**) and the Student's *t*-test (**B, D, E, and I–K**). Data were pooled from two (**B, H, and F**) or three (**A**, C–E, **G, and I–K**) independent experiments.

The online version of this article includes the following figure supplement(s) for figure 2:

**Source data 1.** Source data for *Figure 2A-K*.

**Figure supplement 1.** Expression of CD25, GITR, and BCL2 in tumor-infiltrating Treg cells is independent of CD300a.

**Figure supplement 1—source data 1.** Source data for *Figure 2—figure supplement 1*.

**Figure supplement 2.** Anti-CD25 mAb reduces Treg cells.

*Cd300a*<sup>fl/fl</sup> mice (*Figure 2C*). These Treg cells in *Cd300a*<sup>fl/fl</sup>;*Itgax*<sup>Cre</sup> mice showed higher expression of Ki67 and CTLA-4 than did *Cd300a*<sup>fl/fl</sup> mice, although the expressions of CD25, GITR, and BCL2 on tumor-infiltrating Treg cells were comparable between two genotypes of mice (*Figure 2D and E*, *Figure 2—figure supplement 1*).

To determine whether Treg cells were indeed involved in the exacerbated tumor growth of *Cd300a*<sup>−/−</sup> mice, we depleted Treg cells by using an anti-CD25 monoclonal antibody (mAb) (*Onizuka et al., 1999*; *Figure 2—figure supplement 2*). After Treg cell depletion, the tumor volume of the *Cd300a*<sup>−/−</sup> and *Cd300a*<sup>fl/fl</sup>;*Itgax*<sup>Cre</sup> mice decreased to a level comparable to that seen in wild-type and *Cd300a*<sup>fl/fl</sup> mice, respectively (*Figure 2F and G*). Tumor-infiltrating CD8$^+$ T cells in *Cd300a*<sup>−/−</sup> mice expressed significantly less IFN-γ and significantly higher PD-1 than did those in wild-type mice (*Figure 2H and I*). Moreover, the number of CD8$^+$ T cells expressing both PD-1 and TIM-3 and those expressing CTLA-4 were significantly increased in *Cd300a*<sup>fl/fl</sup>;*Itgax*<sup>Cre</sup> mice compared to *Cd300a*<sup>fl/fl</sup> mice (*Figure 2J and K*), suggesting that tumor-infiltrating CD8$^+$ T cells display more exhausted state in *Cd300a*<sup>fl/fl</sup>;*Itgax*<sup>Cre</sup> mice than in *Cd300a*<sup>fl/fl</sup> mice (*Sawant et al., 2019*). These results suggest that CD300a on DCs regulates the number of tumor-infiltrating Treg cells, which suppress tumor immune responses.

## TEVs augment IFN-β production by DCs

We previously reported that a microbiota-mediated signal induces increased IFN-β production by DCs and increased numbers of Treg cells in the barrier tissues such as the intestine, skin, and airway of *Cd300a*<sup>−/−</sup> mice relative to those of wild-type mice (*Nakahashi-Oda et al., 2016*). In the current study, we found that the expression of *Ifnb* was also higher in DCs in the tumor tissues of *Cd300a*<sup>−/−</sup> and *Cd300a*<sup>fl/fl</sup>;*Itgax*<sup>Cre</sup> mice than in those of wild-type and *Cd300a*<sup>fl/fl</sup> mice, resepectively (*Figure 3A and B*). To examine whether the microbiota is also involved in Treg cell levels in the tumor and tumor growth, we used wild-type and *Cd300a*<sup>−/−</sup> mice raised under the germ-free (GF) conditions. In contrast to the barrier tissues, *Cd300a*<sup>−/−</sup> mice still showed larger numbers of Treg cells and a larger tumor volume than did wild-type mice raised under GF conditions (*Figure 3—figure supplement 1*). These results suggest that, unlike in the barrier tissues, the microbiota-mediated signal was dispensable for the increased numbers of Treg cells in the tumor and for the enhanced tumor growth in *Cd300a*<sup>−/−</sup> mice.

Solid tumors lapse into necrosis in the core region under conditions of hypoxia and low pH, resulting in the secretion of several immune stimulators, such as damage-associated molecular patterns (DAMPs), DNA, RNA (*Patidar et al., 2018*), and EVs (*Couto et al., 2018*). We examined whether the culture supernatant of B16 melanoma cells containing tumor-derived immune mediators had any effect on *Ifnb* expression by using cultured bone marrow-derived dendritic cells (BMDCs). Four hours after incubation in the presence of the culture supernatant, *Cd300a*<sup>−/−</sup> BMDCs expressed higher levels of *Ifnb* than did wild-type BMDCs (*Figure 3C*), suggesting that CD300a suppressed the *Ifnb* expression induced by a tumor-derived immune mediator in the culture supernatant. Since EVs are the particles released from the cells that are delimited by a lipid bilayer that contains phosphatidylserine (*Lima et al., 2009*), the ligand for CD300a (*Nakahashi-Oda et al., 2012a*), and containing

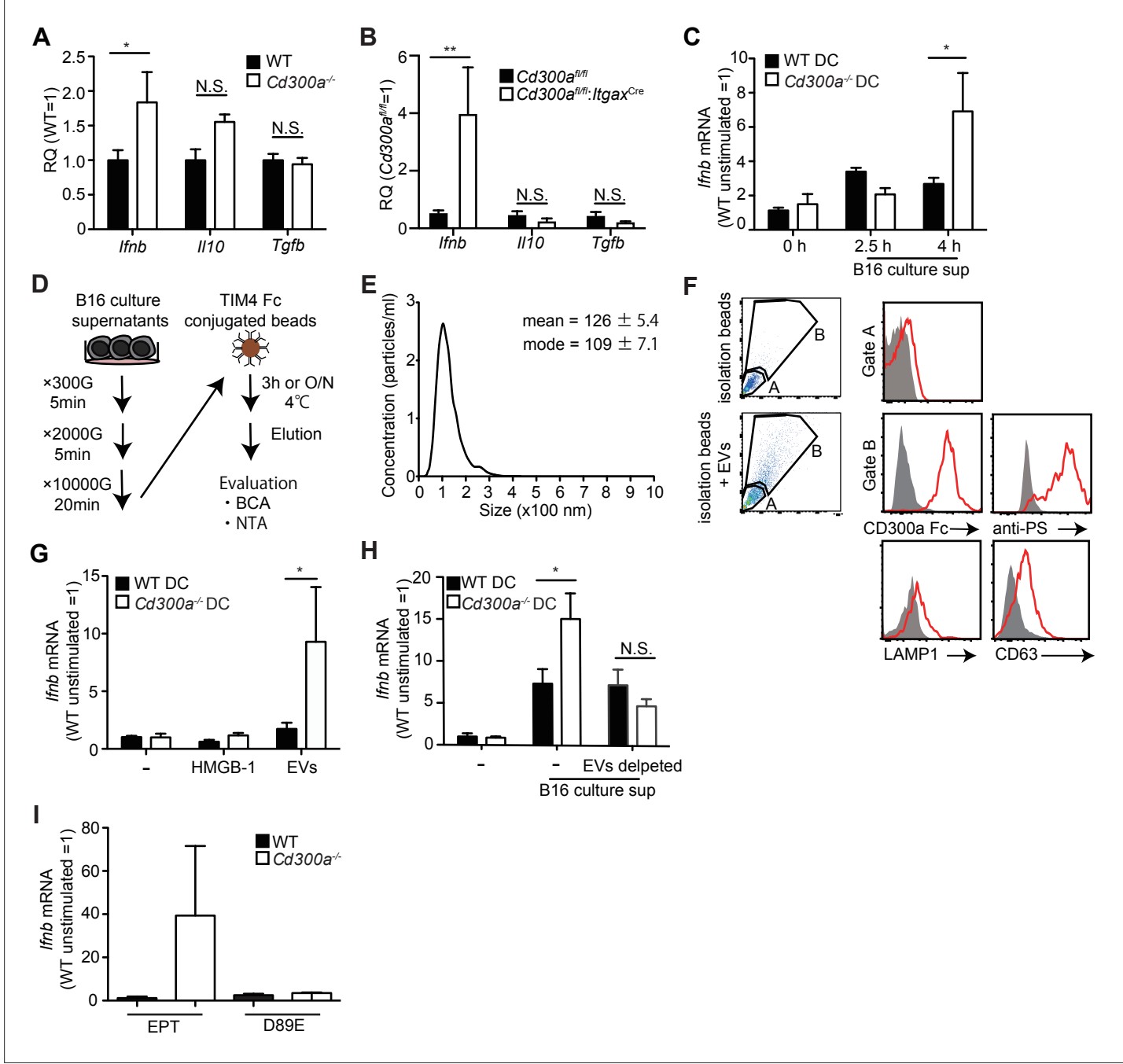

**Figure 3.** Tumor-derived extracellular vesicles (EVs) facilitate interferon-$\beta$ (IFN-$\beta$) production from dendritic cells. (**A and B**) Quantitative everse transcription PCR (RT-PCR) analysis of mRNA from CD11c+ cells sorted from B16 melanoma in wild-type (WT), Cd300afl/fl (n = 6), Cd300a−/− or Cd300afl/fl;ItgaxCre (n = 6) mice 2 weeks after tumor inoculation. Results are presented relative to those of the control gene encoding $\beta$-actin. (**C**) Quantitative RT-PCR analysis of Ifnb in WT- and Cd300a−/−-derived bone marrow-derived dendritic cells (BMDCs) that received no treatment (0 hr, n = 7) or B16 culture supernatants (2.5 hr, n = 5; 4.0 hr, n = 7). (**D**) A schematic illustration of EV isolation. (**E**) The size distribution of isolated B16-derived EVs was analyzed by NTA using NanoSight LM10. (**F**) Flow cytometric analysis of EVs isolated from B16 melanoma supernatants. Bead-conjugated EVs were analyzed by flow cytometry and characterized by the indicated antibody in the presence of 2 mM CaCl₂. (**G and H**) Quantitative RT-PCR analysis of Ifnb in WT and Cd300a−/− BMDCs that received no treatment (–) (n = 6 in each group) and were treated with high-mobility group Box 1 protein (HMGB-1) (n = 3 in each group) or B16-derived EVs (n = 5 in each group) (**G**) or cocultured with B16 cultured supernatant with or without the depletion of EVs (n = 5 in each group) (**H**). (**I**) Quantitative RT-PCR analysis of Ifnb in WT and Cd300a−/− BMDCs that were treated with EPT (control protein; EPT-MFG-E8, n = 3 in each group) or D89E (D89E-MFG-E8, n = 3 in each group). Data are given as means ± standard error of the means (SEMs). RQ: relative quantification; N.S.: not significant. *p < 0.05 and **p < 0.01. p values were obtained by using a two-way analysis of variance (ANOVA) followed by Bonferroni's post-

*Figure 3 continued on next page*

*Figure 3 continued*

test (**A–C, G, and H**). Data were pooled from three (**A–C, G, and H**) independent experiments.

The online version of this article includes the following figure supplement(s) for figure 3:

**Source data 1.** Source data for *Figure 3A-C and G-I*.

**Figure supplement 1.** Tumor growth in *Cd300a⁻/⁻* mice is independent of the microbiota.

**Figure supplement 1—source data 1.** Source data for *Figure 3—figure supplement 1*.

functional biomolecules (*van Niel et al., 2018*). We purified TEVs from the culture supernatants of B16 melanoma cells by centrifugation and phosphatidylserine receptor-conjugated beads (*Figure 3D and E*), which indeed expressed phosphatidylserine on the surface and bound to a chimeric fusion protein of the extracellular portion of CD300a with human IgG1 (*Figure 3F*). Stimulation with the purified TEVs induced higher *Ifnb* expression in *Cd300a⁻/⁻* BMDCs than in wild-type BMDCs (*Figure 3G*). However, *Cd300a⁻/⁻* BMDCs showed decreased *Ifnb* expression to a level comparable to wild-type BMDC when these BMMCs were cultured in the culture supernatants of B16 melanoma cells after removal of TEVs by phosphatidylserine receptor-conjugated beads (*Figure 3H*). In contrast, neither wild-type nor *Cd300a⁻/⁻* BMDCs expressed IFN-β after stimulation high-mobility group box-1 protein (HMGB-1) (*Figure 3G)* a well-known DAMP, which can be released by damaged tumors. These results suggest that the interaction of CD300a with PS on TEVs suppressed *Ifnb* expression in BMDCs. Indeed, *Ifnb* expression in *Cd300a⁻/⁻* BMDCs was decreased to a level comparable to that seen in wild-type BMDCs after stimulation with TEVs whose PS was masked with MFG-E8 protein mutated at residue 89 (D89E-MFG-E8) (*Nakahashi-Oda et al., 2012a*; *Figure 3I*). Together, these results indicated that TEVs suppressed *Ifnb* expression in BMDCs via interaction between CD300a on BMDCs and PS on TEVs.

## TEVs enhanced Treg cell proliferation and consequent tumor development via IFN-β

To clarify whether IFN-β enhances Treg cell proliferation, we cocultured TEV-stimulated wild-type or *Cd300a⁻/⁻* BMDCs with Treg cells that were generated from naive CD4⁺ T cells from Foxp3-eGFP⁺ mice in the presence of anti-CD3 and anti-CD28 mAbs, IL-2, and TGF-β. TEV-stimulated *Cd300a⁻/⁻* BMDCs increased the number of Treg cells to a greater extent than did TEV-stimulated wild-type BMDCs (*Figure 4A*). However, TEV-stimulated *Cd300a⁻/⁻* BMDCs cocultured with Foxp3-eGFP⁺ Treg cells rather than naïve CD4⁺ T cells had the same number of Treg cells as when cocultured with wild-type BMDCs (*Figure 4—figure supplement 1*). Addition of a neutralizing anti-IFN-β antibody to the coculture of Treg cells and *Cd300a⁻/⁻* BMDCs reduced the Treg cell numbers to a level comparable to that seen in the coculture of Treg cells and wild-type BMDCs (*Figure 4A*). Moreover, administration of a neutralizing anti-IFN-β antibody showed reduced tumor volume in *Cd300a*fl/fl;*Itgax*Cre mice to a comparable level of that in *Cd300a*fl/fl mice (*Figure 4B*), suggesting that IFN-β augmented Treg cell proliferation or survival and promoted tumor progression in *Cd300a*fl/fl;*Itgax*Cre mice. To investigate the effects of TEVs on Treg cells, we injected an EV-release inhibitor GW4869 (*Ikebuchi et al., 2018*) into the tumor region on days 10, 14, and 18 after tumor inoculation. Treatment with GW4869, which inhibits EV release from tumors and DCs, led to a significant decrease in the number of tumor-infiltrating Treg cells and the tumor volume in *Cd300a⁻/⁻* and *Cd300a*fl/fl;*Itgax*Cre mice to a comparable level of those in wild-type and *Cd300a*fl/fl mice, respectively (*Figure 4C-E*). These results nevertheless indicate that CD300a suppresses TEV-mediated IFN-β production, resulting in a decrease in the Treg cell population and the suppression of tumor development.

## CD300a inhibits the EV-induced TLR3–TRIF signaling for IFN-β production

To further analyze how CD300a regulates TEV-mediated IFN-β production in DCs, we cocultured pHrodo- or PKH-labeled exosomes with wild-type or *Cd300a⁻/⁻* BMDCs and analyzed the localization of the TEVs in BMDCs by using confocal laser scanning microscopy. We found that the TEVs were incorporated into endosomes, as identified by the expression of endosome antigen (EEA)-1, in both genotypes of DCs (*Figure 5A*). The number of TEVs in the endosomes was comparable between wild-type and *Cd300a⁻/⁻* BMDCs (*Figure 5B*), suggesting that CD300a did not affect TEV incorporation into the endosomes. Interestingly, we also found that CD300a was internalized from the cell

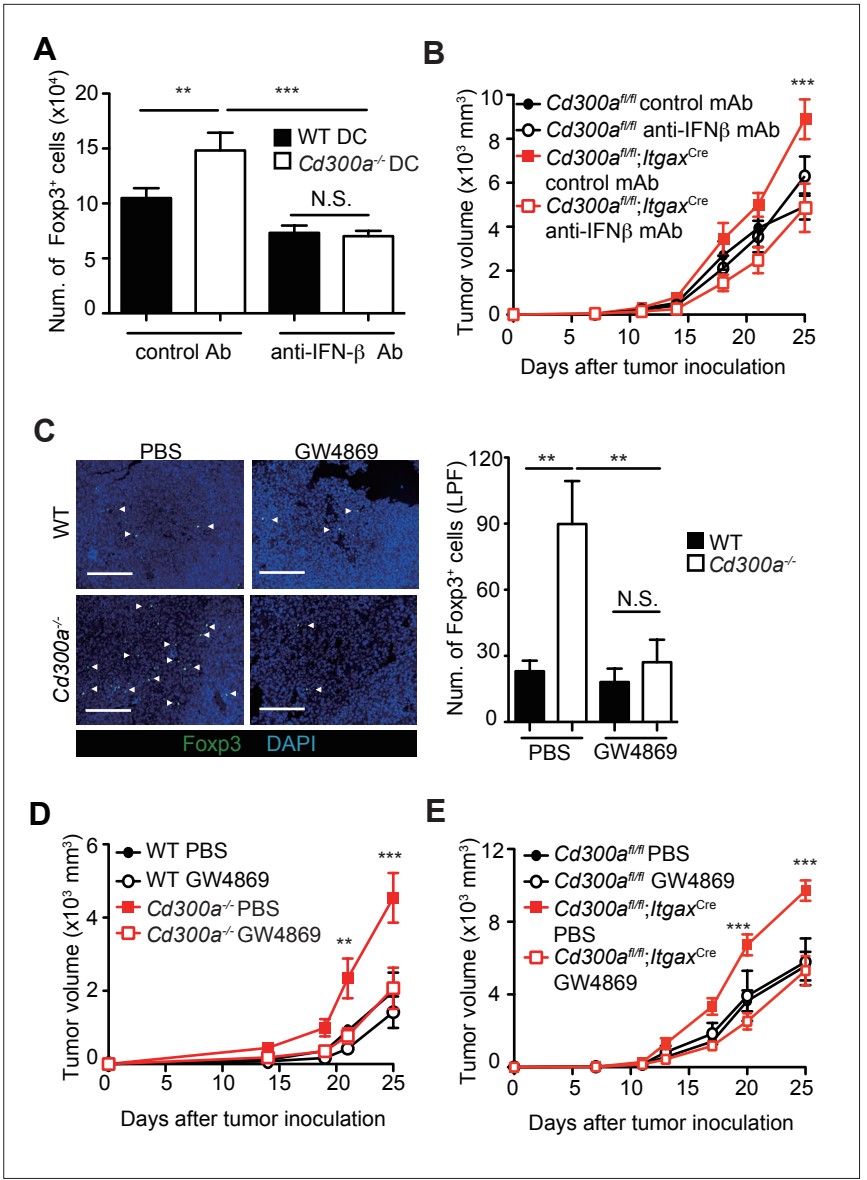

**Figure 4.** Tumor-derived extracellular vesicles (TEVs) promote tumor-infiltrating Treg cell accumulation. (**A**) The number of induced Foxp3-eGFP[+] cells (iTreg) generated from naive T cells by using anti-CD3, anti-CD28, Interleukin-2 (IL-2), and Transforming growth factor-$\beta$ (TGF-$\beta$). These iTreg cells were cocultured with TEV-stimulated bone marrow-derived dendritic cells (BMDCs) in the presence of IL-2 and TGF-$\beta$ for 5 days with a control mAb (n = 7) or an anti-interferon-$\beta$ (IFN-$\beta$) mAb (n = 5). (**B**) Tumor growth curves of *Cd300a*[fl/fl] and *Cd300a*[fl/fl];*Itgax*[Cre] mice treated with an anti-IFN-$\beta$ mAb or a control mAb (control, n = 7; anti-IFN-$\beta$, n = 7 for each genotype mouse) three times (days 7, 11, and 14) and after inoculation of B16 melanoma cells. (**C**) Representative fluorescence micrographs of tumor sections from Foxp3-eGFP wild-type (WT) (phosphate-buffered saline (PBS), n = 4; GW4869, n = 6) and Foxp3-eGFP *Cd300a*[−/−] mice (PBS, n = 5; GW4869, n = 6) in the absence or presence of GW4869, and stained with an anti-GFP mAb (green) and the DNA-binding dye 4′,6-diamidino-2-phenylindole (DAPI, left). The number of Foxp3[+] cells was quantified from four high-power fields (LPF) (right). White arrow shows Foxp3-positive cells. Scale bar, 200 μm. Tumor growth curves of WT (PBS, n = 6; GW4869, n = 9) (**D**) or *Cd300a*[fl/fl] (n = 4 each) (**E**) and *Cd300a*[−/−] mice (PBS, n = 7; GW4869, n = 9) (**D**) or *Cd300a*[fl/fl];*Itgax*-Cre (n = 6 each) (**E**) that were treated with GW4869 or PBS three times (days 14, 18, and 21). Data are given as means ± standard error of the means (SEMs). RQ: relative quantification; N.S.: not significant. **p < 0.01 and ***p < 0.001. p values were obtained by using a one-way analysis of variance (ANOVA) (**A and C**) and a two-way ANOVA followed by Bonferroni's post-test (**B**, **D, and E**). Data were pooled from two (**A and C**) or three (**B, D, and E**) independent experiments.

The online version of this article includes the following figure supplement(s) for figure 4:

*Figure 4 continued on next page*

*Figure 4 continued*

**Source data 1.** Source data for *Figure 4A-E*.

**Figure supplement 1.** CD300a suppression of bone marrow-derived dendritic cells (BMDCs) does not affect the number of Foxp3+ splenic  regulatory T (Treg) cells.

**Figure supplement 1—source data 1.** Source data for *Figure 4—figure supplement 1*.

surface into the endosomes, an event that might be mediated by the tyrosine-based sorting motif in the cytoplasmic region of CD300a (*Yotsumoto et al., 2003*), after coculture of BMDCs with TEVs (*Figure 5C* and *Figure 5—figure supplement 1*). As a result, the TEVs colocalized with CD300a at the endosomes (*Figure 5A and C*). Given that EVs expose phosphatidylserine on their lipid bilayer, which is a CD300a ligand, these results suggest that CD300a was activated via stimulation with TEVs at the endosomes.

EVs also contain nucleic acids, including structured RNA (*Liu et al., 2016*; *van Niel et al., 2018*). TLR3 at the endosomal membrane can recognize RNA and mediates IFN-β production via the TRIF signaling pathway in DCs (*Tatematsu et al., 2013*). To examine whether CD300a inhibited TLR3-mediated signaling at the endosomes upon stimulation with TEVs, we cocultured wild-type and *Cd300a*⁻/⁻ BMDCs with TEVs in the presence of an inhibitor of TLR3 (*Cheng et al., 2011*). This inhibitor decreased *Ifnb* expression in *Cd300a*⁻/⁻ BMDCs to a level comparable to that in wild-type BMDCs (*Figure 5D* and *Figure 5—figure supplement 2*). In contrast, the TLR4 inhibitor TKA-242 did not affect the expression of *Ifnb* in either BMDC genotype (*Figure 5D* and *Figure 5—figure supplement 2*). These results suggest that CD300a inhibits TLR3-mediated signaling for IFN-β production. Moreover, the expression of *Ifnb* in *ticam-1*⁻/⁻;*Cd300a*⁻/⁻ BMDCs was also decreased to the comparable level of that in *ticam-1*⁻/⁻ BMDCs after coculture with TEVs (*Figure 5E*). In addition, we found that the phosphorylation level of interferon regulatory factor 3 (IRF3), a downstream molecule of the TRIF signaling pathway, was increased to a greater extent in EV-stimulated *Cd300a*⁻/⁻ BMDCs than in wild-type BMDCs (*Figure 5F*). In vivo analyses also showed that, although tumor growth was significantly larger and the survival rate was significantly shorter for B16 melanoma-injected *Myd88*⁻/⁻;*Cd300a*⁻/⁻ mice compared with B16-injected *Myd88*⁻/⁻ mice, tumor development and survival did not differ between *ticam-1*⁻/⁻;*Cd300a*⁻/⁻ and *ticam-1*⁻/⁻ mice (*Figure 5G-J*). Taken together, these data suggest that CD300a inhibits the TLR3–TRIF signaling pathway for IFN-β production at the endosomes in DCs, resulting in the suppression of Treg cell activation and tumor development.

## CD300A expression associates with survival times in melanoma patients

To examine the role of CD300A in tumor development in humans, we analyzed the data on the single-cell RNA sequence (scRNA-seq) of human melanoma tissues, which demonstrated that *CD300A* is expressed on populations that express *HLA-DR*, *ITGAX (CD11C)*, *ITGAM (CD11B)*, *CD14*, and *CD163* (*Figure 6—figure supplement 1*), consistent with the results of mouse melanoma. We further analyzed the database of the Cancer Genome Atlas (TCGA) project and found that skin cutaneous melanoma (SKCM) patients expressing low levels of *CD300A* mRNA had shorter survival times than did those expressing higher *CD300A* mRNA levels (*Figure 6A*). We also found that the expression ratio of *CD300A* to *ITGAX* is negatively correlated with that of *FOXP3* to *CD8A* (*Figure 6B*). These results suggested that CD300A suppressed Treg cell proliferation and/or activation and tumor development. Moreover, we found that patients with melanoma showed strong positive correlation between *FOXP3* and *IFNB1* expression (*Figure 6C*). Neutral sphingomyelinase-2 (SMPD3), which is a target of an inhibitor of EV-release GW4869, enhances TEV release from tumor cells (*Kosaka et al., 2013*; *Kosaka et al., 2010*). TCGA database of SKCM also showed a strong positive correlation between expressions of *SMPD3* and *IFNB1* in melanoma tissues (*Figure 6D*), suggesting that TEVs increased IFN-β expression in human melanoma tissues. These results were consistent with those of mouse models of melanoma development in the current study. Taken together, these results suggested that CD300A might augment tumor immunity via suppression of tumor-infiltrating Treg cells also in humans.

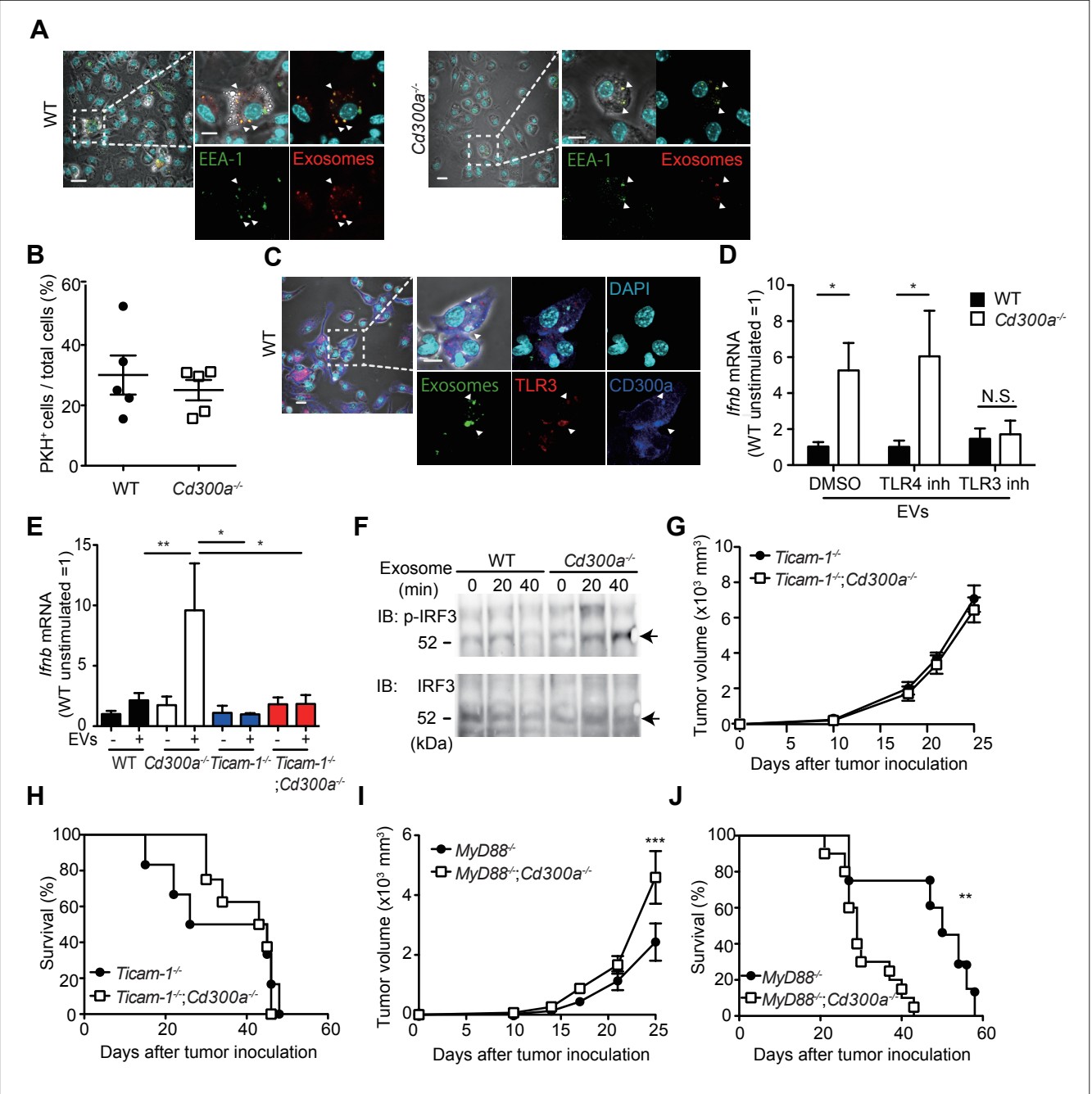

**Figure 5.** CD300a inhibits TLR3-mediated interferon- $\beta$ (IFN- $\beta$ ) expression upon recognition of tumor-derived exosomes. (**A**) Representative microscopy images of wild-type (WT) and $Cd300a^{-/-}$ bone marrow-derived dendritic cells (BMDCs) treated with pHrodo-labeled extracellular vesicles (EVs) to assess the localization of EVs (red) and early endosome antigen (EEA)-1 (green). Scale bar, 10 µm. Data are representative of two independent experiments. (**B**) Uptake of PKH-labeled tumor-derived EVs (TEVs) in WT (*n* = 5) and $Cd300a^{-/-}$ BMDCs (*n* = 5). (**C**) Representative microscopy images of WT and $Cd300a^{-/-}$ BMDCs treated with pHrodo-labeled exosomes to assess the localization of exosomes (green), TLR3 (red), and CD300a (blue). Scale bar, 10 µm. Data are representative of two independent experiments. (**D**) Quantitative RT-PCR analysis of *Ifnb* in WT and $Cd300a^{-/-}$ BMDCs treated with B16-derived exosomes in the presence of dimethyl sulfoxide (DMSO) (WT, *n* = 9; $Cd300a^{-/-}$, *n* = 10), 100 nM TLR4 inhibitor (*n* = 7 in each group), and 50 µM TLR3 inhibitor (*n* = 6 in each group). (**E**) Quantitative RT-PCR analysis of *Ifnb* in WT, $Cd300a^{-/-}$, ticam-1$^{-/-}$, and ticam-1$^{-/-}$;$Cd300a^{-/-}$ mice-derived BMDCs treated with B16-derived EVs (*n* = 5 in all group). (**F**) Representative immunoassay of WT and $Cd300a^{-/-}$ BMDCs left unstimulated (0 min) or stimulated for the indicated times with B16-derived exosomes, followed by immunoblot analysis of phosphorylated (p-) interferon regulatory factor 3 (IRF3) or total IRF3. Data are representative of two independent experiments. (**G and H**) Comparison of tumor growth and survival curves of B16 melanoma cells between *ticam-1*$^{-/-}$ (*n* = 6) and *ticam-1*$^{-/-}$;$Cd300a^{-/-}$ ice (*n* = 9) after inoculation of B16 melanoma. (**I and J**) Comparison of tumor growth and survival curves of B16 melanoma between *MyD8*$^{-/-}$ (*n* = 9) and *MyD88*$^{-/-}$;$Cd300a^{-/-}$ mice (*n* = 10) after inoculation of B16 melanoma. Data

*Figure 5 continued on next page*

*Figure 5 continued*

are given as means ± standard error of the means (SEMs). N.S.: not significant. *p < 0.05, **p < 0.01, and ***p < 0.001. p values were obtained by using the Student's *t*-test (**B**), a two-way analysis of variance (ANOVA) followed by Bonferroni's post-test (**D, E, G, and I**), and the log-rank test (**H and J**). Data were pooled from two (**B, E, and H**) or three (**D, I, and J**) independent experiments.

The online version of this article includes the following figure supplement(s) for figure 5:

**Source data 1.** Source data for *Figure 5B, D, E, and G-J*.

**Figure supplement 1.** CD300a is localized on the surface of plasma membrane without stimulation.

**Figure supplement 2.** The inhibitors of TLR4 and TLR3 suppress the expression of *Ifnb*.

**Figure supplement 2—source data 1.** Source data for *Figure 5—figure supplement 2*.

## Discussion

Although the biological roles of EVs have been reported from various angles, how EVs regulate immune responses is not yet fully understood. In the present study, we showed that TEV-stimulated DCs for IFN-β production via TLR3 at the endosomes, resulting in the increased number of tumor-infiltrating Treg cells and thus the exacerbation of tumor development. In contrast, the TEVs also stimulated CD300a and inhibited TEV-mediated TLR3 signaling at the endosome. Thus, TEVs have both positive and negative functions in the regulation of IFN-β production and Treg activation via the axis of EV-derived RNA-TLR3 and EV-derived phosphatidylserine-CD300a, respectively. These results suggest that the Treg cells in tumor microenvironments is regulated by the balance of positive and negative signaling for IFN-β production induced by TEV (*Figure 7*). Hence, it is an interesting issue to be examined whether the expressions of RNAs are different among TEVs derived from tumors of variable tissue types.

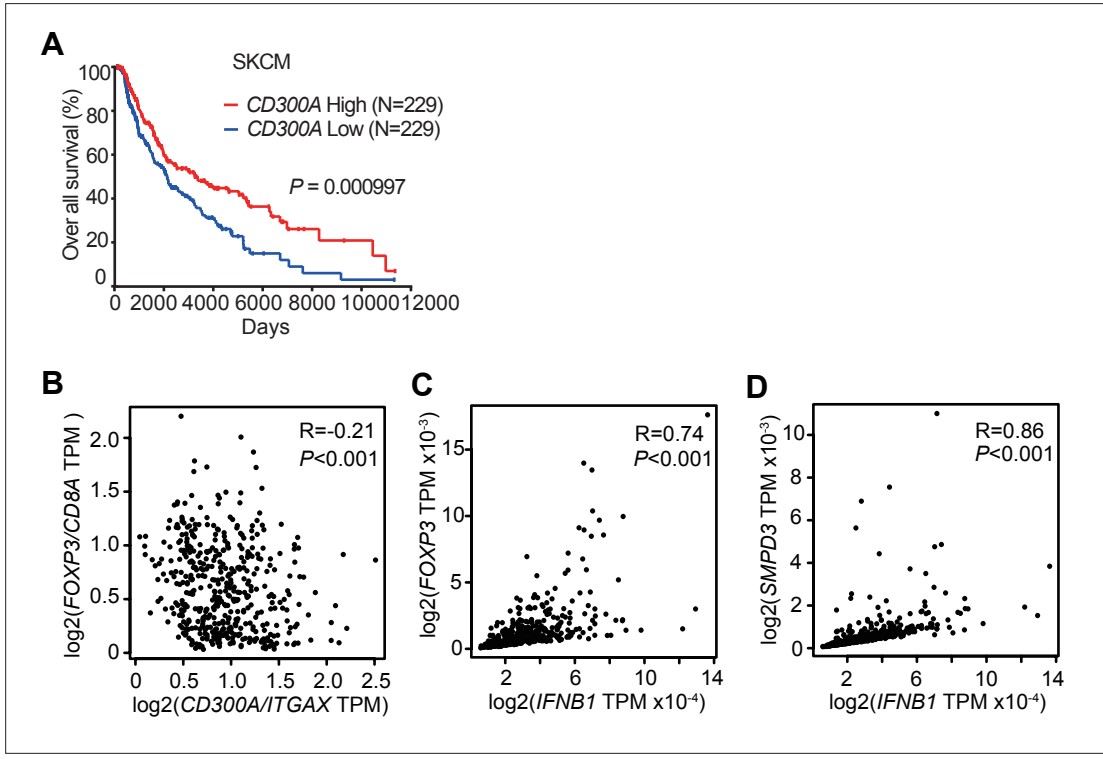

**Figure 6.** CD300A expression associates with survival of human melanoma patients. (**A**) Kaplan plot showing low and high *CD300A* expressions in skin cutaneous melanoma (SKCM) patients obtained by performing a meta-analysis of The Cancer Genome Atlas (TCGA) database. Median values were used as thresholds (numbers of both low and high expression patients = 229). (**B–D**) Spearman correlation analysis of TCGA skin cutaneous melanoma database by using GEPIA2. *FOXP3, IFNB1,* and *SMPD3* expression were normalized by *GAPDH* expression (**C and D**).

The online version of this article includes the following figure supplement(s) for figure 6:

**Figure supplement 1.** *t*-Distributed stochastic neighbor embedding (tSNE) plots of the immune cell landscape isolated from melanoma patients.

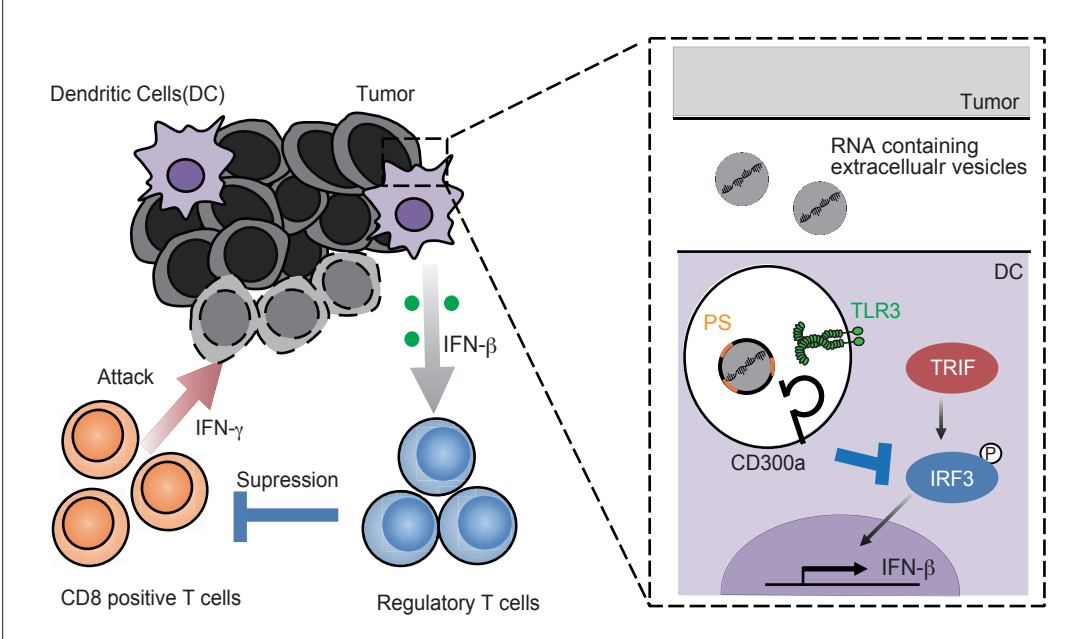

**Figure 7.** A schematic model of the role of tumor-derived extracellular vesicle (TEV) and CD300a in tumor immunity. RNA-containing TEV is incorporated into the endosomes in dendritic cells (DCs), where phosphatidylserine (PS) on TEV binds to CD300a and inhibits the TLR3–TRIF–IRF3 signaling pathway initiated by TEV-derived RNA binding to TLR3, resulting in the decrease in interferon-$\beta$ (IFN-$\beta$) production by DCs and the number of tumor-infiltrating Treg cells. The Treg cells regulate tumor development.

On the other hand, the balance of TLR3 and CD300a expressions in DCs may also be important for Treg activation and tumor development. Indeed, we showed that higher expression of CD300A was associated with lower expression of Foxp3 and longer survival times of melanoma patients. Though human CD300A is reported to be expressed on the subsets of T cells in peripheral blood (*Borrego, 2013*), human melanoma tissues showed quite low CD300A expression in T cells compared to high expression in the myeloid cells. Further analysis of human CD300A expressions on T cells in tumor microenvironment is needed. While previous reports demonstrated that TEVs promoted Treg cell expansion through DC-independent manner in vitro (*Muller et al., 2017*; *Szajnik et al., 2010*; *Wieckowski et al., 2009*), the current study first demonstrated that TEVs regulate Treg cell activation and tumor development in vivo by DCs in the tumor microenvironment. Meanwhile, Tumor-infiltrating DCs are heterogenous, and can be divided into at least two subsets. The conventional type-1 DC (cDC1) expresses the chemokine receptor XCR1 and CD103 and lower amount of CD11b that has the high ability to migrate from tumors to lymph nodes and presents a tumor antigen to CD8[+] T cells (*Bedoui et al., 2009*). In contrast, the conventional type-2 DC (cDC2) are commonly distinguished from cDC1 by their preferential expression of higher amount of CD11b. cDC2 are predominantly involved in antigen presentation by MHC class II to CD4[+] T cells (*Gao et al., 2013*). Given that cDC2 is involved in CD4[+] T cell differentiation and activation, CD300a on cDC2, rather than cDC1, may regulate Treg cells activation by inhibiting the TLR3–IFN-β pathway in tumor microenvironment.

Type I IFNs are key players in antiviral and anticancer immune response by upregulating both cross-presentation of antigens by CD8a[+] DCs and cytotoxic activity of CD8[+] T cells and NK cells (*Zitvogel et al., 2015*). However, the current clinical use of IFN-β for cancers showed limited efficiency (*Medrano et al., 2017*; *Minn, 2015*). This may be partly due to the fact that type I IFNs also have immunosuppressive functions, such as inducing the production of IL-10, an immunosuppressive cytokine (*Snell et al., 2017*). In addition, on Treg cells, IFNs are most potent cytokines to induce PD-L1 (*Morimoto et al., 2018*; *Xiao et al., 2018*), which contributes to sustain Foxp3 expression and promotes the function of Treg cells (*Francisco et al., 2009*). In addition, IFN-α/β receptor signaling promotes Treg cell development (*Metidji et al., 2015*). We previously reported that gut commensals stimulated CX3CR1[+]CD103[−] CD11b[+] DCs to produce IFN-β, which augmented the proliferation of Treg cells in the intestine (*Nakahashi-Oda et al., 2016*). In contrast, published reports demonstrated

that, in viral infection and tumor microenvironment, type I IFNs directly inhibit the proliferation and activation of Treg cells (*Gangaplara et al., 2018*; *Srivastava et al., 2014*). Further investigations are required to clarify whether such IFN-β-induced Treg cell increase is caused by Treg cell-intrinsic or Treg cell-extrinsic effect by the cytokine in the tumor microenvironment.

We have previously reported that CD300a inhibited the CD14-mediated TLR4 internalization in CD11b⁺ DCs induced by gut microbiota (*Nakahashi-Oda et al., 2016*). This internalization of TLR4 by CD14 induces activation of the TRIF pathway to produce IFN-β, but not MyD88 pathway. In the present study, we also found that CD300a inhibits TLR3-mediated TRIF signaling to produce IFN-β (*Figure 5D and E*). These data suggest that CD300a specifically inhibits TRIF signaling to produce IFN-β, rather than the MyD88-mediated signaling pathway stimulated by LPS and HMGB-1. TLR3 activates PI3 kinase and the downstream kinase, Akt, leading to full phosphorylation and activation of IRF3 (*Sarkar et al., 2004*). Indeed, we showed that IRF3 phosphorylation was increased in *Cd300a⁻/⁻* DCs compared with wild-type DCs after stimulation with TEV. Recent studies have revealed that TLR3 on alveolar epithelial cells recognized RNAs in TEV and promoted lung metastasis (*Liu et al., 2016*). Therefore, the role of RNAs in TEV is dependent on target cells. Our findings thus highlighted the role of TEV and CD300a on DCs in the regulation of tumor-infiltrating Treg cells and tumor immunity.

# Materials and methods

**Key resources table**

| Reagent type (species) or resource | Designation | Source or reference | Identifiers | Additional information |
|---|---|---|---|---|
| Genetic reagent (*M. musculus*) | *Cd300a⁻/⁻* | PMID:26855029 | | |
| Genetic reagent (*M. musculus*) | *Cd300a*fl/fl | PMID:26855029 | | |
| Genetic reagent (*M. musculus*) | *Cd300a*fl/fl;*Lyz2*Cre | PMID:26855029 | | |
| Genetic reagent (*M. musculus*) | *Cd300a*fl/fl;*Itgax*Cre | PMID:26855029 | | |
| Genetic reagent (*M. musculus*) | *Ticam1⁻/⁻;Cd300a⁻/⁻* | PMID:26855029 | | |
| Genetic reagent (*M. musculus*) | *MyD88⁻/⁻;Cd300a⁻/⁻* | PMID:26855029 | | |
| Genetic reagent (*M. musculus*) | *Rag1⁻/⁻;Cd300a⁻/⁻* | PMID:26855029 | | |
| Genetic reagent (*M. musculus*) | *Ticam1⁻/⁻* | PMID:1285581 Oriental Bio Service | | |
| Genetic reagent (*M. musculus*) | *MyD88⁻/⁻* | PMID:9697844 Oriental Bio Service | | |
| Genetic reagent (*M. musculus*) | *Rag1⁻/⁻* | Jackson Laboratory | | |
| Genetic reagent (*M. musculus*) | *Foxp3*eGFP | PMID:18209052 | | Dr. B. Malissen (UM2 Aix-Marseille Université) |
| Genetic reagent (*M. musculus*) | *Foxp3*eGFP;*Cd300a⁻/⁻* | PMID:26855029 | | |
| Cell line (*M. musculus*) | B16 | RIKEN Cell Bank | RCB1283 RRID:CVCL_ F936 | |
| Antibody | Rat anti-CD8-PE-Cy7 (53–6.7, mouse monoclonal) | BD Bioscience | Cat# 552,877 RRID:AB_ 394,506 | FACS (1:5) |
| Antibody | Rat anti-CD4-APC (RM4-5, mouse monoclonal) | BD Bioscience | Cat# 553,051 RRID:AB_398528 | FACS (1:5) |

*Continued on next page*

*Continued*

| Reagent type (species) or resource | Designation | Source or reference | Identifiers | Additional information |
|---|---|---|---|---|
| Antibody | Rat anti-CD11b-APC-Cy7 (M1/70, monoclonal) | BD Bioscience | Cat# 557,657 RRID:AB_396772 | FACS (1:5) |
| Antibody | Armenian hamster anti-CD11c FITC (HL3, monoclonal) | BD Bioscience | Cat# 553,801 RRID:AB_553801 | FACS (1:5) |
| Antibody | Rat anti-I-A/I-E BV500 (M5/114.15.2, monoclonal) | BD Bioscience | Cat# 562,366 RRID:AB_11153488 | FACS (1:5) |
| Antibody | Rat anti-Ly6G PE (1A8, monoclonal) | BD Bioscience | Cat 551,461 RRID:AB_394208 | FACS (1:5) |
| Antibody | Rat anti-CD62L PE (MEL-14, monoclonal) | BD Bioscience | Cat# 553151, RRID:AB_394666 | FACS (1:5) |
| Antibody | Rat anti-CD44 APC (IM7, monoclonal) | BD Bioscience | Cat# 559250, RRID:AB_398661 | FACS (1:5) |
| Antibody | Rat anti-CD25 PE (PC61, monoclonal) | BD Bioscience | Cat# 553866, RRID:AB_395101 | FACS (1:5) |
| Antibody | Mouse anti-Ki67 Alexa Fluor 647(B56, monoclonal) | BD Bioscience | Cat# 558615, RRID:AB_647130 | FACS (1:5) |
| Antibody | Rat anti-IFN-g Alexa Fluor 488 (XMG1.2, monoclonal) | BD Bioscience | Cat# 557724, RRID:AB_396832 | FACS (1:5) |
| Antibody | Rat anti-CD63 APC-Cy7 (NVG-2, monoclonal) | BioLegend | Cat# 143907, RRID:AB_2565497 | FACS (1:5) |
| Antibody | Armenian hamster anti-CD103 APC (2E7, monoclonal) | BioLegend | Cat# 121,414 RRID:AB_1227502 | FACS (1:5) |
| Antibody | Mouse anti-XCR1 PE (ZET, monoclonal) | BioLegend | Cat# 148,204 RRID:AB_2563843 | FACS (1:5) |
| Antibody | Mouse anti-Bcl-2 PE (BCL/10C4, monoclonal) | BioLegend | Cat# 633508, RRID:AB_2290367 | FACS (1:5) |
| Antibody | Rat anti-GITR PE-Cy7 (DTA-1, monoclonal) | BioLegend | Cat# 126317, RRID:AB_2563385 | FACS (1:5) |
| Antibody | Rat anti-PD-1 PE-Cy7 (RMP1-30, monoclonal) | BioLegend | Cat# 109,109 RRID:AB_572016 | FACS (1:5) |
| Antibody | Rat anti-Tim3 PE (RMT3-23, monoclonal) | BioLegend | Cat# 119703, RRID:AB_345377 | FACS (1:5) |
| Antibody | Armenian hamster anti-CTLA-4 biotin (UC10-4B9, monoclonal) | BioLegend | Cat# 106303, RRID:AB_313252 | FACS (1:5) |
| Antibody | Mouse anti-Foxp3 Alexa Fluor 488 (150D, monoclonal) | BioLegend | Cat# 320012, RRID:AB_439748 | FACS (1:5) |
| Antibody | Mouse anti-CD45.2 APC (104, monoclonal) | BioLegend | Cat# 109814, RRID:AB_389211 | FACS (1:5) |
| Antibody | Rat anti-CD40 APC (3/23, monoclonal) | BioLegend | Cat# 124612, RRID:AB_1134072 | FACS (1:5) |
| Antibody | Rat anti-IFNβ (7F-D3, monoclonal) | Yamasa | Cat# 7,891 | Vivo (50 µg/mouse, three times) |
| Antibody | Mouse anti-PS FITC (1H6, monoclonal) | Merck Millipore | Cat# 16-256, RRID:AB_492616 | FACS (1:5) |
| Antibody | Rat anti-CD16/CD32 (2.4G2, monoclonal) | TONBO Bioscience | Cat# 70-0161, RRID:AB_2621487 | FACS (1:5) |
| Antibody | Mouse anti-CD300a (EX42, monoclonal) | PMID:31155312 | | FACS (0.1 µg) |

*Continued on next page*

*Continued*

| Reagent type (species) or resource | Designation | Source or reference | Identifiers | Additional information |
|---|---|---|---|---|
| Antibody | Rabbit anti-GFP (D5.1, monoclonal) | Cell signaling | Cat# 2956, RRID:AB_1196615 | IHC (1:200) |
| Antibody | Rat anti-Foxp3 (FJK-16s, monoclonal) | Thermo Fisher | Cat#14-5773-82 RRID:AB_467576 | IHC (1:200) |
| Antibody | Mouse anti-EEA-1 (1G11, monoclonal) | eBioscience | Cat# 14-9114 RRID:AB_2572929 | ICC (1:200) |
| Antibody | Rat anti-TLR3 (11F8, monoclonal) | Biolegend | Cat# 141902, RRID:AB_10901162 | ICC (1:200) |
| Antibody | Rabbit anti-phosphorylated IRF3 (4D4G, monoclonal) | Cell Signaling Technology | Cat# 4,947 | WB (1:1000) |
| Antibody | Rabbit anti-IRF3 (FL-425, polyclonal) | Santa Cruz Biotechnology | Cat# sc-9082, RRID:AB_2264929 | WB (1:1000) |
| Antibody | Hamster anti-CD3 purified (145–2 C11, monoclonal) | TONBO Bioscience | Cat# 70-0031, RRID:AB_2621472 | Cell culture (0.33 µg/ml) |
| Antibody | Syrian hamster anti-CD28 purified (37.51, monoclonal) | Biolegend | Cat# 102101, RRID:AB_312866 | Cell culture (2 µg/ml) |
| Sequence-based reagent | Ifnb_F | This paper | PCR primers | CAGCTCCAAGAAAGGACGAAC |
| Sequence-based reagent | Ifnb_R | This paper | PCR primers | GGCAGTGTAACTCTTCTGCAT |
| Sequence-based reagent | Il10_F | This paper | PCR primers | GCTGGACAACATACTGCTAACC |
| Sequence-based reagent | Il10_R | This paper | PCR primers | ATTTCCGATAAGGCTTGGCAA |
| Sequence-based reagent | Tgfb_F | This paper | PCR primers | TGACGTCACTGGAGTTGTACGG |
| Sequence-based reagent | Tgfb_R | This paper | PCR primers | GGTTCATGTCATGGATGGTGC |
| Peptide, recombinant protein | Streptavidin | Thermo Fisher | Cat. #: 434,302 | |
| Peptide, recombinant protein | GM-CSF | WAKO | Cat. #: 434,302 | |
| Peptide, recombinant protein | IL-4 | WAKO | Cat. #: 434,302 | |
| Peptide, recombinant protein | IL-2 | BD Pharmingen | | |
| Peptide, recombinant protein | TGF-β | R&D system | | |
| Commercial assay or kit | Exosome Isolation Kit | WAKO | | |
| Commercial assay or kit | Tumor dissociation kit | Miltenyi Biotec | | |
| Commercial assay or kit | High-Capacity cDNA Reverse Transcription Kit | Applied Biosystems | | |
| Commercial assay or kit | Power SYBER Green PCR Master Mix | Applied Biosystems | | |
| Chemical compound, drug | GW4869 | Cayman Chemial | Cat. #: 13,127 | |

*Continued on next page*

*Continued*

| Reagent type (species) or resource | Designation | Source or reference | Identifiers | Additional information |
|---|---|---|---|---|
| Chemical compound, drug | TLR3/dsRNA complex inhibitor | Merck | Cat. #: 614,310 | |
| Chemical compound, drug | TLR4 inhibitor (TAK-242) | Merck | Cat. #: 614,316 | |
| Chemical compound, drug | pHrodo Red ester | Thermo Fisher | | |
| Chemical compound, drug | pHrodo STP Green | Thermo Fisher | | |
| Chemical compound, drug | HRP-conjugated dextran polymer | PerkinElmer | | |
| Software, algorithm | GraphPad Prism | GraphPad Prism | | |
| Software, algorithm | Hybrid cell counts software | Keyence | | |
| Software, algorithm | R Seurat | R: The R Project for Satistical Computing | | |
| Software, algorithm | FlowJo | TreeStar | | |

## Mice

All gene-edited mice in the C57BL/6 J background were previously described (*Nakahashi-Oda et al., 2016*). C57BL6J mice and GF mice were purchased from Clea Japan and Sankyo Laboratory, respectively. GF mice were bred and maintained in vinyl isolators to maintain GF conditions. Mice were used for the experiments at 8–12 weeks of age. All experiments were performed in accordance with the guidance of the animal ethics committee of the University of Tsukuba Animal Research Center.

## Antibodies, flow cytometry, and reagents

The isotype-matched control antibodies rat IgG2a (553928), rat IgG1 (553921), and mouse IgG1 (553445), as well as mAbs to CD4 (RM4-5), CD8 (53–6.7), CD11b (M1/70), CD11c (HL3), I-A/I-E (M5/114.15.2), Ly6C (AL-21), Ly6G (1A8), CD62L (MEL-14), CD44 (IM7), CD25 (PC61), Ki67 (B56), and IFN-γ (XMG1.2) were purchased from BD Bioscience. Mabs to CD63 (NVG-2), CD103 (2E7), XCR1 (ZET), Bcl-2 (BCL/10C4), GITR (DTA-1), PD-1 (RMP1-30), Tim3 (RMT3-23), CTLA-4 (UC10-4B9), Foxp3 (150D), CD45.2 (104), and CD40 (3/23) were purchased from Biolegend. Anti-IFN-β(7F-D3) was from Yamasa; control rat IgG (6130-01) was purchased from Southern Biotechnology. Anti-PS antibody (1H6) was purchased from Merck Millipore. The CD300a-specific mAb (EX42) was generated in our laboratory. Anti-CD25 (PC61) was a gift from E. Nakayama (Okayama University). Cells were treated for 10 min with anti-CD16/CD32 mAb (2.4G2; TONBO Bioscience) to prevent binding to FcγR prior to incubation with the indicated combination of antibodies. All samples were evaluated by using a Fortessa flow cytometer (Becton Dickinson) and analyzed by using FlowJo software (Tree Star).

## Tumor cell maintenance and injection

The B16 mouse melanoma cell line was obtained from RIKEN Cell Bank (Tsukuba, Japan). Authentication and mycoplasma contamination test (DNA staining method, polymerase chain reaction [PCR method]) were also performed at RIKEN Cell Bank (Tsukuba, Japan). Identification of mouse strain (Simple sequence length polymorphism analysis), identification of animal species (PCR method), morphology, cell viability, and adhesion efficiency had performed for authentication according to ICLAC. Cells were maintained in RPMI-1640 (Sigma) supplemented with 5 % (vol/vol) fetal bovine serum (FBS) (Thermo Fisher). To inoculate the tumor cells into mice, cells were harvested by trypsinization, washed with sterile PBS, and injected intradermally ($2 \times 10^5$ cells/50 µl sterile PBS/mouse) on the flank of each mouse. Tumor growth was measured every three or 4 days by using a caliper.

## Cell preparations

For tumor-infiltrating Treg cell preparation, tumor tissues were harvested 3 weeks after tumor inoculation. Tumor tissues were cut into small pieces, incubated in 5 % FBS RPMI-1640 in the presence of an enzyme mixture (Miltenyi Biotec) at 37°C for 45 min, and digested by using a gentleMACS Dissociator and tumor dissociation kit (Miltenyi Biotec), according to the manufacturer's instructions. Cells were filtered through 70 μm nylon mesh and subsequently centrifuged using different concentrations of Percoll (Sigma-Aldrich) to exclude tissue debris and were washed with staining medium.

BMDCs were generated as described previously (*Nakahashi-Oda et al., 2016*). Briefly, bone marrow cells were cultured in a 10 cm culture dish in complete RPMI-1640 containing 10 % FBS in the presence of 10 ng/ml GM-CSF (WAKO) and 10 ng/ml IL-4 (WAKO) for 7 days. BMDCs were enriched by using CD11c MACS Beads (Miltenyi Biotec) to remove dead cells generated during BMDC development.

## Cytokine production from tumor-infiltrating lymphocytes

Cells were isolated from tumors in mice 3 weeks after inoculation, and stimulated for 4 hr with 50 ng/ml PMA and 500 ng/ml ionomycin. Brefeldin A (Sigma-Aldrich) was added for the last 3 hr of culture. Cells were treated by using Foxp3 staining kits (eBioscience) and then stained with anti-IFN-γmAb.

## Immunohistochemistry and immunocytochemical staining

Paraffin-embedded tumor samples were deparaffinized in xylene and a series of graded concentrations of alcohol. To block endogenous horseradish peroxidase (HRP), tissue sections were incubated in 0.3 % hydrogen peroxidase in methanol for 30 min at room temperature. For antigen retrieval, the specimens were preheated in AR6 buffer (PerkinElmer). Samples were incubated with anti-GFP (D5.1) XP (Cell signaling) or Rat anti-Foxp3 (FJK-16s; Thermo Fisher) for 1 hr at room temperature or overnight at 4 °C, respectively, and then incubated with appropriate secondary HRP-conjugated Abs. An HRP-conjugated dextran polymer system (PerkinElmer) was used for detection. After being washed with TBST, sections were mounted with 4′,6-diamidino-2-phenylindole (DAPI; Vector labs). For quantification of Foxp3+ cells in tumor tissues, tissue sections were scanned using BZ-X710 (Keyence). The number of Foxp3+ cells per high-power field in each area was automatically counted with hybrid cell counts software (Keyence). For immunocytochemical staining, $1.0 \times 10^5$ BMDCs were cultured in eight-well chamber slides (Thermo Fisher) and were stimulated with pHrodo Red ester or pHrodo STP Green (Thermo Fisher)-labeled exosomes. Cells were then fixed with 10 % paraformaldehyde at 4 °C for 20 min, permeabilized with 0.3 % Triton-X, and then stained with a mAb to EEA-1 (1G11; eBioscience) or TLR3 (11F8; Biolegend), followed by Alexa Flor 488-conjugated donkey anti-mouse IgG or Alexa Flor 546-conjugated goat anti-rat IgG (Invitrogen), respectively. Samples were evaluated by use of laser scanning confocal microscopy (FV10i FLOUVIEW; Olympus).

## In vivo depletion of Treg cells

For in vivo depletion of Treg cells, mice were injected intraperitoneally with 300 μg of an anti-CD25 mAb (PC61) and an isotype control Ab on days −6, −3, and 0 before B16 tumor inoculation.

## EV inhibitor treatment

To inhibit EV generation, mice were injected with 1.0 mg/kg GW4869 (*Ikebuchi et al., 2018*; *Kosaka et al., 2013*) (Cayman Chemical) intratumorally on days 14, 18, and 21 after tumor inoculation. Tumor tissues were harvested on day 25.

## EV isolation and treatment

B16 melanoma cells were cultured in complete RPMI supplemented with or without 2 % bovine serum albumin. The culture medium was harvested and subjected to sequential centrifugation steps (first, 5 min for 2000 G; second, 20 min for 10,000 G). EVs were purified by using an Exosome Isolation Kit (WAKO) according to the manufacturer's protocol. In brief, streptavidin magnetic beads, bound with biotinylated mouse Tim4-Fc, which is the phosphatidylserine receptors, were added to the culture medium of B16 melanoma containing 2 mM CaCl$_2$, and the mixture was rotated for 3 hr or overnight at 4 °C. The beads were washed three times with washing buffer and exosomes were eluted with elution buffer (*Figure 3D*). For quantification of the EVs in the elution buffer, the concentration of EV protein was quantified by using a BCA Protein Assay Kit (Novagen). For BMDC stimulation by EVs, 2

× 10$^5$ BMDCs were incubated in the presence of 3–5 μg/ml EVs for 2.5 h. To inhibit TLR3 and TLR4 signaling, a TLR3/dsRNA complex inhibitor (Merck) and a TLR4 inhibitor (TAK-242; Merck) were added to the cultures of BMDCs for 15 min before exosome stimulation.

## Coculture of iTreg cells with EV-stimulated BMDCs

CD4+ T cells were enriched from the spleen cells by using mouse CD4 MACS Beads (L3T4, Miltenyi Biotec) and then CD4$^+$CD44$^{lo}$CD62L$^{high}$Foxp3-eGFP$^−$ naive T cells were purified by sorting with flow cytometry (FACS Aria III, Becton Dickinson). Inducible Treg cells were generated by culture of naive CD4$^+$ T cells in the presence of plate-coated 0.33 μg/ml anti-CD3 Ab (145–2 C11; TONBO), 2.0 μg/ml soluble CD28 (37.51; Biolegend), 20 ng/ml IL-2 (BD Pharmingen), and 2.5 ng/ml TGF-β (R&D system) for 3 days. Inducible Treg cells (5 × 10$^4$ cells/well) were cultured with exosome-stimulated BMDCs (5 × 10$^4$ cells/well) in 96-well round-bottom plates in the presence of IL-2 and TGF-β for 5 days.

## Quantitative real-time PCR analysis

Total RNA was extracted from tumor-infiltrating CD11c + cells and BMDCs. Reverse transcription was performed with a High-Capacity cDNA Reverse Transcription Kit (Applied Biosystems). Quantitative PCR analysis was performed with Power SYBER Green PCR Master Mix (Applied Biosystem) by using an ABI 7500 sequence detector (Applied Biosystems). The PCR primers are as follows: *Ifnb* fwd, 5′-cagctccaagaaaggacgaac-3′; *Ifnb* rev, 5′-ggcagtgtaactcttctgcat-3′; *Il10* fwd, 5′-gctggacaacatactgctaacc-3′; *Il10* rev, 5′- atttccgataaggcttggcaa-3′; and *Tgfb* fwd, 5′-tgacgtcactggagttgtacgg-3′; *Tgfb* rev, 5′-ggttcatgtcatggatggtgc-3′; normalization of quantitative real-time PCR was performed based on the gene encoding β-actin.

## Western blots

BMDCs were stimulated or unstimulated with exosomes for 20 or 40 min and lysed with 1% NP-40. The lysetes of BMDCs were immunoblotted with antibody to phosphorylated IRF3 (4D4G; Cell Signaling Technology) or IRF3 (FL-425; Santa Cruz Biotechnology).

## Bioinformatics

For analysis of melanoma scRNA-seq, data were downloaded from the database of scRNA-seq analysis of melanoma (accession no. GSE72056). The matrix data were passed to the R software package Seurat. Cells that had unique gene counts of <200 were excluded, as were all genes that were expressed in >3 cells. Counted data were log2-transformed and scaled by Seurat's *Scale Data* function. Principal component (PC) analysis was performed on a set of highly variable genes defined by Seurat's *FindVariableGenes* function. Genes associated the resulting PCs were then used for dimensionality reduction by using *t*-distributed stochastic neighbor embedding. Cluster-based marker identification and differential expression were performed using Seurat's *FindAllmarkers*. RNA-seq and survival data were obtained from TCGA project and analyzed by using OncoLnc and GEPIA (*Anaya, 2016*; *Tang et al., 2017*).

## Statistical analyses

Comparisons were performed using GraphPad Prism version 5.0 (GraphPad Software) by one- or two-way analysis of variance, followed by Bonferroni's multiple comparisons test or Student's unpaired *t*-test. Data are presented as means ± standard error of the means, and differences are considered significant at p < 0.05.

## Acknowledgements

We thank Hisako Furugen, Satoko Tochihara, and Wakako Saito for secretarial assistance. This research was supported in part by grants provided by the Ministry of Education, Culture, Sports, Science, and Technology of Japan (grant numbers 18H05022 and 16H06387 to AS and 19H03776 and 16H05350 to CN-O) and a grant-in-aid from the Japan Society for the Promotion of Science Fellows (grant number 17J06167 to YN).

## Additional information

### Funding

| Funder | Grant reference number | Author |
|---|---|---|
| Japan Society for the Promotion of Science | 19H03776 | Chigusa Nakahashi-Oda |
| Japan Society for the Promotion of Science | 18H05022 | Akira Shibuya |
| Japan Society for the Promotion of Science | 16H05350 | Chigusa Nakahashi-Oda |
| Japan Society for the Promotion of Science | 16H06387 | Akira Shibuya |
| Japan Society for the Promotion of Science | 17J06167 | Yuta Nakazawa |

The funders had no role in study design, data collection and interpretation, or the decision to submit the work for publication.

### Author contributions

Yuta Nakazawa, Formal analysis, Investigation, Methodology, Writing - original draft; Nanako Nishiyama, Hitoshi Koizumi, Investigation; Kazumasa Kanemaru, Methodology, Supervision, Validation; Chigusa Nakahashi-Oda, Conceptualization, Funding acquisition, Investigation, Methodology, Project administration, Writing - original draft, Writing - review and editing; Akira Shibuya, Funding acquisition, Supervision, Validation, Writing - original draft, Writing - review and editing

### Author ORCIDs

Kazumasa Kanemaru https://orcid.org/0000-0003-0018-8966
Chigusa Nakahashi-Oda https://orcid.org/0000-0003-2288-3628
Akira Shibuya https://orcid.org/0000-0002-4480-4858

### Ethics

This study was performed in strict accordance with the recommendations in the Guide for the Care and Use of Laboratory Animals of the National Institutes of Health. All of the animals were handled according to approved institutional animal care and use committee (IACUC) protocols (#08-133) of the University of Arizona. The protocol was approved by the Committee on the Ethics of Animal Experiments of the University of Tsukuba Animal Research Center (Permit Number:19-231). All surgery was performed under isoflurane anesthesia, and every effort was made to minimize suffering.

### Decision letter and Author response

Decision letter https://doi.org/10.7554/eLife.61999.sa1
Author response https://doi.org/10.7554/eLife.61999.sa2

## Additional files

### Supplementary files

• Transparent reporting form

### Data availability

All data generated or analysed during this study are included in the manuscript and supporting files. Source data files have been provided for Figures 1 to 5 and figure supplements for Figures 2 to 5.

The following previously published datasets were used:

| Author(s) | Year | Dataset title | Dataset URL | Database and Identifier |
|---|---|---|---|---|
| Tirosh I, Izar B, Prakadan S, Wadsworth M, Treacy D, Trombetta J, Rotem A, Rodman C, Lian C, Murphy G, Fallahi-Sichani M, Dutton-Regester K, Lin J, Cohen O, Shah P, Lu D, Genshaft A, Shalek AK, Regev A, Garraway LA | 2016 | Single cell RNA-seq analysis of melanoma | https://www.ncbi. nlm.nih.gov/geo/ query/acc.cgi?acc= GSE72056 | NCBI Gene Expression Omnibus, GSE72056 |

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
