## [Editor Report]

This report shows that the inhibitory immunoreceptor CD300a binding tumor-derived extracellular vesicles are incorporated into dendritic cells and inhibit their IFN-β production in tumor tissues. This results in suppressed activation of tumor-infiltrating regulatory T cells and consequently enhanced tumor immunity.

---

## [Decision Letter]

**Decision letter after peer review:**

Thank you for submitting your article "Tumor-derived extracellular vesicles regulate tumor-infiltrating regulatory T cells via the immunoreceptor CD300a" for consideration by *eLife*. Your article has been reviewed by 3 peer reviewers, including a member of our Board of Reviewing Editors, and the evaluation has been overseen by Tadatsugu Taniguchi as the Senior Editor. The following individual involved in review of your submission has agreed to reveal their identity: Licia Rivoltini (Reviewer #3).

The reviewers have discussed the reviews with one another and the Reviewing Editor has drafted this decision to help you prepare a revised submission.

As the editors have judged that your manuscript is of interest, but as described below that additional experiments are required before it is published, we would like to draw your attention to changes in our revision policy that we have made in response to COVID-19 (https://elifesciences.org/articles/57162). First, because many researchers have temporarily lost access to the labs, we will give authors as much time as they need to submit revised manuscripts. We are also offering, if you choose, to post the manuscript to bioRxiv (if it is not already there) along with this decision letter and a formal designation that the manuscript is "in revision at eLife". Please let us know if you would like to pursue this option. (If your work is more suitable for medRxiv, you will need to post the preprint yourself, as the mechanisms for us to do so are still in development.)

Summary:

Using a mouse model of melanoma, this report demonstrates the relevance of the CD300a immunoreceptor, specifically in dendritic cells (DCs), in tumor growth. It shows that the absence of CD300a is correlated with a higher number of regulatory T cells (Tregs) within the tumor microenvironment and therefore the tumor grows faster and survival decreases. Based on additional experiments, the authors propose a mechanism by which tumor-derived extracellular vesicles (TEVs) interact with CD300a in DCs, decreasing IFNbeta production which subsequently reduces the number of Tregs. In addition, data from melanoma patients show a correlation between overall survival and higher levels of CD300a expression in the tumor.

Essential revisions:

1. It is highly recommended to clearly demonstrate the role of IFNbeta in the proposed mechanism. In addition to using an anti-IFNbeta mAb in an in vitro culture (Figure 3D), other experiments must be performed, such as in vivo experiments with the anti-IFNbeta mAb. The authors have used this mAb in their previously published article (Nakahashi-Oda et al., Nature Immunology, 2016). Alternatively, in vivo experiments could also be performed with IFNAR1-like (IFN α and β receptor 1 subunit) KO animals.

In addition, is the observed increase in Tregs within the tumor in CD300a-/- animals due only to an increase in IFNbeta production by DCs? Are there other cytokines and/or cell-cell contact that may play a role? At least this should be discussed.

2. Why are not all the experiments performed on CD300afl/fl Itgax-Cre mice instead of CD300a-/- mice? The experiments in Figures 2a, S2C, 3 and 4 should have been performed on CD300afl/fl Itgax-Cre mice. This is very important to state unequivocally that only CD300a in DCs is involved in the induction of an immune response capable of inhibiting tumor development.

3. The authors found expansion of tumor-infiltrating Tregs in mice deficient in CD300a. However, no increase in Tregs was observed in tumor-draining lymph nodes. Did authors assess the expression of Treg activation and proliferation molecular markers, such as CD25, CTLA4, GITR, CD39, CD73 or Ki67? If indeed, Treg expansion as a result of CD300a-deficiency is the cause of enhanced tumor growth, authors should provide more evidence of Treg suppressive response. For example, authors can consider measuring the levels of co-stimulatory molecules (e.g. CD40, CD80 and CD86) on dendritic cells, which generally correlate with Treg activitiy and/or tumoral IL-2 concentration.

4. PD-1 is the only marker analyzed to assess the exhausted status of CD8^+^ T cells infiltrating tumor lesion of CD300a-/- mice. Additional evidence of this functional status could be provided, such as for instance expression of CTLA4, TIM3 or other immune checkpoints, or low Ki67 levels. Indeed, particularly in reference of the human setting, PD-1 is also a sign of T cell activation, usually expressed in T cells infiltrating highly immunogenic and hot tumors. Hence, it would be useful having a broader characterization of immune effectors associated with progressing tumor microenvironment when CD300a is lost.

5. Since authors have Foxp3-reporter mice, they should confirm their data in Figure 3D with natural / freshly isolated Tregs, unless they are suggesting that CD300a mainly prevents in situ conversion of intra-tumoral CD4^+^Foxp3- Tconv cells into Tregs.

6. Given that the interaction between CD300a and phosphatidylserine (PS) is critical to CD300a activation, PS co-localization with CD300a ought to be included in confocal microscopy. In addition, the binding of CD300a to PS and PE, which are both upregulated in dead cells, implies that apoptotic bodies could also be shuttling comparable signaling, Can the authors exclude that these particles are present in the EV preparations? Furthermore, does tumor supernatant lose any effect when depleted of EVs? The latter evidence could significantly strengthenthe exclusive involvement of exosomes in the process.

7. Did authors validate the importance of PS in the context that they propose with an anti-PS blocking antibody? There are not many anti-PS blocking antibodies available and they might not block engagement with CD300a (see Nat Commun. 2016 Mar 14;7:10871). Nonetheless, this would be a good assay to demonstrate PS as the ligand that triggers CD300a to inhibit TLR3 and subsequent IFN-β production.

---

## [Author Response]

Essential revisions:1. It is highly recommended to clearly demonstrate the role of IFNbeta in the proposed mechanism. In addition to using an anti-IFNbeta mAb in an in vitro culture (Figure 3D), other experiments must be performed, such as in vivo experiments with the anti-IFNbeta mAb. The authors have used this mAb in their previously published article (Nakahashi-Oda et al., Nature Immunology, 2016). Alternatively, in vivo experiments could also be performed with IFNAR1-like (IFN α and β receptor 1 subunit) KO animals.In addition, is the observed increase in Tregs within the tumor in CD300a-/- animals due only to an increase in IFNbeta production by DCs? Are there other cytokines and/or cell-cell contact that may play a role? At least this should be discussed.

According to the reviewer’s comment, we administrated anti-IFN-β mAb or control Ab into tumor-bearing *Cd300a*^fl/fl^;*Itgax*-Cre and control *Cd300a*^fl/fl^ mice to elucidate the in vivo role of IFN-β in CD300a-deficient mice. We showed that whereas the tumor size was comparable between *Cd300a*^fl/fl^ mice that received control mAb a comparable level to those mice treated with control mAb. We included these results in Figure 4B and described in line 204 – 206 in the revised manuscript. These in vivo data together with in vitro data (Figure 4A) have provided the evidence that IFN-β from DCs expressing CD300a is involved in tumor promotion in CD300a-deficient mice.

However, as reviewer suggested, it remains possible that in addition to IFN-β, other cytokines or DC-Treg cell contact in CD300a-deficient DCs are also involved in the increase of Treg cell number. We discussed this point in line 331 – 333 in the revised manuscript.

2. Why are not all the experiments performed on CD300afl/fl Itgax-Cre mice instead of CD300a-/- mice? The experiments in Figures 2a, S2C, 3 and 4 should have been performed on CD300afl/fl Itgax-Cre mice. This is very important to state unequivocally that only CD300a in DCs is involved in the induction of an immune response capable of inhibiting tumor development.

As suggested by the reviewer, the experiments in Figure 2A, Figure 3A and Figure 3E in submitted manuscript were also performed using *Cd300a*^fl/fl^ and *Cd300a*^fl/fl^;*Itgax*-Cre mice. These results were shown in Figure 2C, Figure 3B and Figure 4E in revised manuscript, respectively. Furthermore, we added the new experimental results using *Cd300a*^fl/fl^ and *Cd300a*^fl/fl^;*Itgax*-Cre mice in Figures 2D, 2E, 2J, 2K and 3B, and Figure 2—figure supplement 1. On the other hand, Figure 3—figure supplement 1 used germ free mice, and Figures 5E and 5G – J used double knockout mice. Since it takes long time to prepare such mice on the *Cd300a*^fl/fl^ and *Cd300a*^fl/fl^;*Itgax*-Cre background, we did not perform these experiments using *Cd300*a conditional mice.

3. The authors found expansion of tumor-infiltrating Tregs in mice deficient in CD300a. However, no increase in Tregs was observed in tumor-draining lymph nodes. Did authors assess the expression of Treg activation and proliferation molecular markers, such as CD25, CTLA4, GITR, CD39, CD73 or Ki67? If indeed, Treg expansion as a result of CD300a-deficiency is the cause of enhanced tumor growth, authors should provide more evidence of Treg suppressive response. For example, authors can consider measuring the levels of co-stimulatory molecules (e.g. CD40, CD80 and CD86) on dendritic cells, which generally correlate with Treg activitiy and/or tumoral IL-2 concentration.

According to the reviewer’s suggestion, we further analyzed the activation and proliferation markers of Treg cells and found that the expression of Ki67 and CTLA-4 were upregulated in tumor-bearing *Cd300a*^fl/fl^;*Itgax*-Cre mice compared with *Cd300a*^fl/fl^ mice (Figure 2D and ２E), although the expressions of CD25, GITR, and BCL2 were comparable between two genotypes of mice (Figure 2—figure supplement 1). Furthermore, we found that the increase in PD-1^+^ TIM3^+^ (exhausted) CD8^+^ T cells in *Cd300a*^fl/fl^;*Itgax*-Cre mice (Figure 2J) and the decrease in IFN-β production from CD8^+^ T cells (Figure 2H). In addition, Treg cell depletion suppressed tumor development in *Cd300a^-/-^* and *Cd300a*^fl/fl^;*Itgax*-Cre mice to a level comparable to that in wild-type and *Cd300a*^fl/fl^ mice, respectively (Figure 2F and 2G). Together, these results provided the evidence that activation and proliferation of Treg cell promoted tumor growth in *Cd300a^-/-^* and *Cd300a*^fl/fl^;*Itgax*-Cre mice.

It is unclear why no increase in Tregs was observed in tumor-draining lymph nodes.

However, since CD300a is highly expressed on conventional DC2 (CD11c^hi^ CD11b^+^ CD103^-^ XCR1^-^ DCs) (Figure 1E), which is known to show a low ability to migrate to lymph nodes as we discussed in line 309-312 (Gao et al., 2013), it is likely that CD300a on DCs may regulate Treg cell proliferation in the tumor microenvironment, rather than in the draining lymph nodes, by increased IFN-β production by DC.

We described these results in line 305 – 315.

4. PD-1 is the only marker analyzed to assess the exhausted status of CD8^+^ T cells infiltrating tumor lesion of CD300a-/- mice. Additional evidence of this functional status could be provided, such as for instance expression of CTLA4, TIM3 or other immune checkpoints, or low Ki67 levels. Indeed, particularly in reference of the human setting, PD-1 is also a sign of T cell activation, usually expressed in T cells infiltrating highly immunogenic and hot tumors. Hence, it would be useful having a broader characterization of immune effectors associated with progressing tumor microenvironment when CD300a is lost.

Besides PD-1 expression, we further evaluated the expressions of TIM3 and CTLA-4 on CD8^+^ T cells. We found that the percentage of PD-1^+^ TIM3^+^ CD8^+^ T cells and the expression of CTLA-4 were elevated in *Cd300a*^fl/fl^;*Itgax*-Cre mice compared to *Cd300a*^fl/fl^ mice (Figure 2J and K). These data suggest that CD8^+^ T cells were much exhausted in *Cd300a*^fl/fl^;*Itgax*-Cre mice than those in *Cd300a*^fl/fl^ mice. We described these results in line 137 – 141 (with citing reference).

5. Since authors have Foxp3-reporter mice, they should confirm their data in Figure 3D with natural / freshly isolated Tregs, unless they are suggesting that CD300a mainly prevents in situ conversion of intra-tumoral CD4^+^Foxp3- Tconv cells into Tregs.

As suggested by the reviewer, Foxp3+ Treg cells (nTreg cells) isolated from spleen were co-cultured with EV-stimulated DCs from wild-type and *Cd300a^-/-^* mice for analysis. However, the number of Treg cells was comparable between DCs from wild-type and *Cd300a^-/-^* mice (Figure 4—figure supplement 1). Thus, our data (Figure 4A) showed that IFN-β derived from *Cd300a^-/-^* DCs may promote the proliferation or survival of inducible Treg cells in the tumor microenvironment. We described these results in line 194 -201.

6. Given that the interaction between CD300a and phosphatidylserine (PS) is critical to CD300a activation, PS co-localization with CD300a ought to be included in confocal microscopy. In addition, the binding of CD300a to PS and PE, which are both upregulated in dead cells, implies that apoptotic bodies could also be shuttling comparable signaling, Can the authors exclude that these particles are present in the EV preparations? Furthermore, does tumor supernatant lose any effect when depleted of EVs? The latter evidence could significantly strengthenthe exclusive involvement of exosomes in the process.

As pointed out by the reviewer, the interaction of CD300a and phosphatidylserine (PS) is critical for CD300a activation. However, the EVs themselves are too small to be detected by confocal microscopy; their average size is 126 nm, as shown in Figure 3E. Instead, we showed in the original version of the manuscript the binding of CD300a to the PS on the bead-bound EV by means of flow cytometry (Figure 3F in revised manuscript) and endocytosed EV by confocal microscopy (Figure 5C).

We agree with the reviewer that dead cells, but not EVs, possibly bound CD300a and mediate inhibitory signal in DCs. To exclude this possibility, EVs were purified by two successive centrifugations to remove dead cells and macroscopic debris followed by magnetic beads, finally yielding only one-peak of EVs around 126 nm (Figure 3D and E). Furthermore, when DCs were cultured in the tumor supernatant after removal of EVs, IFN-β expression in *Cd300a-/-* DCs was reduced to a level comparable to that in wild-type DCs (Figure 3H), indicating that EVs, rather than dead cells, were involved in the CD300a-mediated suppression of IFN-β expression in DCs. We described these results in line 172-180.

7. Did authors validate the importance of PS in the context that they propose with an anti-PS blocking antibody? There are not many anti-PS blocking antibodies available and they might not block engagement with CD300a (see Nat Commun. 2016 Mar 14;7:10871). Nonetheless, this would be a good assay to demonstrate PS as the ligand that triggers CD300a to inhibit TLR3 and subsequent IFN-β production.

We agree with the reviewer that there are few anti-PS antibody and they may not block the binding of PS to CD300a. Therefore, to clarify that PS is a ligand for CD300a, we used mutated MFG-E8 protein (D89E-MFG-E8), which specifically binds to PS and blocks PS interaction with CD300a (Nakahashi-Oda et al., BBRC, 2012a). When D89E-MFG-E8 protein was added to the co-culture of exosomes and DCs, IFN-β expression in *Cd300a^-/-^* DCs decresed to a level found in wild-type BMDCs (Figure 3I). We described these results in line 185 – 188.